



# Description and evaluation of aerosol in UKESM1 and HadGEM3-GC3.1 CMIP6 historical simulations

Jane P. Mulcahy[1], Colin Johnson[1,2], Colin G. Jones[2], Adam C. Povey[3], Catherine E. Scott[4], Alistair Sellar[1], Steven T. Turnock[1], Matthew T. Woodhouse[5], N. Luke Abraham[6,11], Martin B. Andrews[1], Nicolas Bellouin[7], Jo Browse[4,a], Ken S. Carslaw[4], Mohit Dalvi[1], Gerd A. Folberth[1], Matthew Glover[1], Daniel Grosvenor[4], Catherine Hardacre[1], Richard Hill[1], Ben Johnson[1], Andy Jones[1], Zak Kipling[8,b], Graham Mann[4], James Mollard[7], Fiona M. O'Connor[1], Julien Palmieri[9], Carly Reddington[4], Steven T. Rumbold[10], Mark Richardson[4], Nick A. J. Schutgens[8,c], Philip Stier[8], Marc Stringer[10], Yongming Tang[1], Jeremy Walton[1], Stephanie Woodward[1], and Andrew Yool[9]

[1]Met Office Hadley Centre, Exeter, UK, EX1 3PB
[2]National Centre for Atmospheric Science, University of Leeds, UK
[3]National Centre for Earth Observation, University of Oxford, OX1 3PU, UK
[4]School of Earth and Environment, University of Leeds, UK
[5]CSIRO Climate Science Centre, Aspendale, Victoria, Australia
[6]National Centre for Atmospheric Science, University of Cambridge, UK
[7]Department of Meteorology, University of Reading, UK
[8]Atmospheric, Oceanic and Planetary Physics, Department of Physics, University of Oxford, UK
[9]National Oceanography Centre, Southampton, UK
[10]National Centre for Atmospheric Science, University of Reading, UK
[11]Department of Chemistry, University of Cambridge, Cambridge, UK, CB2 1EW
[a]now at: University of Exeter, Penryn, UK
[b]now at: ECMWF, Shinfield Park, Reading, RG2 9AX, UK
[c]now at: Earth Sciences, Faculty of Science, Vrije Universiteit Amsterdam, Amsterdam, the Netherlands

**Correspondence:** J. P. Mulcahy (jane.mulcahy@metoffice.gov.uk)

**Abstract.** We document and evaluate the aerosol schemes as implemented in the physical and Earth system models, HadGEM3-GC3.1 (GC3.1) and UKESM1, which are contributing to the 6th Coupled Model Intercomparison Project (CMIP6). The simulation of aerosols in the present-day period of the historical ensemble of these models is evaluated against a range of observations. Updates to the aerosol microphysics scheme are documented as well as differences in the aerosol representation between the physical and Earth system configurations. The additional Earth-system interactions included in UKESM1 leads to differences in the emissions of natural aerosol sources such as dimethyl sulfide, mineral dust and organic aerosol and subsequent evolution of these species in the model. UKESM1 also includes a stratospheric-tropospheric chemistry scheme which is fully coupled to the aerosol scheme, while GC3.1 employs a simplified aerosol chemistry mechanism driven by prescribed monthly climatologies of the relevant oxidants. Overall, the simulated speciated aerosol mass concentrations compare reasonably well with observations. Both models capture the negative trend in sulfate aerosol concentrations over Europe and the eastern United States of America (US) although the models tend to underestimate the sulfate concentrations in both regions. Interactive emissions of biogenic volatile organic compounds in UKESM1 lead to an improved agreement of organic aerosol over the US.





Simulated dust burdens are similar in both models despite a 2-fold difference in dust emissions. Aerosol optical depth is biased low in dust source and outflow regions but performs well in other regions compared to a number of satellite and ground-based retrievals of aerosol optical depth. Simulated aerosol number concentrations are generally within a factor of 2 of the observations with both models tending to overestimate number concentrations over remote ocean regions, apart from at high latitudes,

and underestimate over Northern Hemisphere continents. Finally, UKESM1 includes for the first time a representation of a primary marine organic aerosol source. The impact of this new aerosol source is evaluated. Over the pristine Southern Ocean, it is found to improve the seasonal cycle of organic aerosol mass and cloud droplet number concentrations relative to GC3.1 although underestimations in cloud droplet number concentrations remain. This paper provides a useful characterization of the aerosol climatology in both models facilitating the understanding of the numerous aerosol-climate interaction studies that will

be conducted as part of CMIP6 and beyond.

# 1 Introduction

Atmospheric aerosols are an important component of the Earth system due to their impacts on the radiation characteristics of the atmosphere and on cloud and precipitation processes. Aerosol particles scatter and absorb incoming solar and outgoing thermal radiation modifying the radiation balance of the atmosphere (Haywood and Boucher, 2000; Ramanathan et al., 2001; Forster et al., 2007; Boucher et al., 2013). Aerosols also act as cloud condensation nuclei on which cloud droplets can form.

Increasing aerosol concentrations due to increased anthropogenic emissions leads to increases in cloud droplet number concentrations ($N_d$) enhancing cloud albedo (Twomey, 1977) and may also modify precipitation frequency and distribution (Albrecht, 1989; Lohmann and Feichter, 2005). In the Earth system aerosols influence and are influenced by atmospheric chemistry and biogeochemical cycles in the atmosphere and on land, ocean, and ice surfaces (Mercado et al., 2009; Carslaw et al., 2010; Mahowald, 2011). Global climate and Earth system models have therefore historically attempted to represent aerosol-cloud-

radiation interactions with varying levels of realism (Penner et al., 2001; Ghan and Schwartz, 2007; Boucher et al., 2013). Aerosols remain one of the largest uncertainties in the latest estimates of anthropogenic radiative forcing on climate (Boucher et al., 2013) and aerosol feedbacks within the Earth system are often neglected.

The large uncertainty in aerosol forcing on climate is due primarily to the large uncertainites associated with aerosol-cloud interactions, how we represent these processes in models (Ghan et al., 2016; Gryspeerdt et al., 2017) and how we

calculate changes in cloud properties over the industrial period (Carslaw et al., 2013). The sub-grid scale nature of these interactions makes accurately simulating the underpinning processes in global climate models (GCMs) inherently difficult,





potentially impacting our ability to accurately simulate historical and future climate change (Shindell et al., 2013; Rotstayn et al., 2015; Bodas-Salcedo et al., 2019). In addition, despite the increasing complexity of aerosol models, uncertainties in aerosol emissions, chemical processing, optical properties and removal rates compound uncertainties in estimates of aerosol effective radiative forcing (Regayre et al., 2018; Karset et al., 2018; Lee et al., 2013). An unprecedented number of global

model simulations of past and future climate change are being conducted as part of the 6th Climate Model Intercomparison Project (CMIP6, Eyring et al. (2016)) and offer a new opportunity to explore and quantify aerosol-climate interactions using the latest state-of-art global climate and Earth System models (ESMs), many of which now include more advanced aerosol schemes compared to that used in CMIP5.

The first version of the United Kindgom Earth System Model, UKESM1 (Sellar et al., 2019a) is the latest Earth sytem model
developed jointly by the UK's Met Office and Natural Environment Research Council (NERC) and will contribute significantly to CMIP6. UKESM1 is built on top of the Global Coupled 3.1 (GC3.1) configuration of the HadGEM3 (Hadley Centre Global Environment Model) physical climate model (Kuhlbrodt et al., 2018; Williams et al., 2017). The physical model is commonly referred to as "HadGEM3-GC3.1" but we shall hereafter refer to it as GC3.1. UKESM1, in addition, includes important ocean and land biogeochemical processes (representing the global carbon cycle) which can have amplifying or damping feedbacks
on physical climate change and/or change themselves in response to changes in the physical climate. UKESM1 also includes the full stratospheric-tropospheric chemistry scheme (Archibald et al., 2019) implemented as part of the United Kingdom Chemistry and Aerosol (UKCA) model, which is coupled to the chemical production of sulphate and secondary organic aerosol. Dynamic vegetation and interactive ocean biogeochemical processes are coupled to natural aerosol emissions of dust, dimethyl sulphide (DMS) and marine and terrestrial organic compounds. Mineral dust is deposited to the ocean and as a source of soluble
iron influences ocean productivity. The UKESM1 and GC3.1 models provide an ideal opportunity to explore aerosol forcing and feedbacks in a traceable hierarchy of models which vary in Earth System process complexity and therefore feedbacks. A detailed characterization of the aerosol properies in both models is therefore essential to underpin and facilitate understanding in such studies.

The aerosol scheme implemented in both UKESM1 and GC3.1 represents a step-change in complexity compared to the mass-
based, bulk aerosol scheme, CLASSIC (Coupled Large-scale Aerosol Simulator for Studies In Climate), used in the preceeding Earth System model, HadGEM2-ES (Jones et al., 2011; Collins et al., 2011). Both UKESM1 and GC3.1 employ the modal version of the Global Model of Aerosol Processes (GLOMAP-mode, hereafter referred to as GLOMAP) 2-moment, aerosol microphysics scheme (Mann et al., 2010) for all aerosol species apart from mineral dust which still uses the bin scheme of Woodward (2001). Bellouin et al. (2013) compared both CLASSIC and GLOMAP aerosol schemes in a previous configuration
of HadGEM3 and found a weaker aerosol-cloud albedo interaction but a stronger aerosol-radiation interaction in GLOMAP and highlighted the potential for improved aerosol forcing estimates with the advanced modal treatment of aerosol and aerosol microphysics. However, more recent investigations by Mulcahy et al. (2018) found an overly strong, negative aerosol effective radiative forcing of -2.77 $\mathrm{Wm^{-2}}$ from 1850 to the year 2000 in an atmosphere-only configuration of HadGEM3-GC3.0 which also employs the GLOMAP scheme. They highlighted a number of issues in the underlying physical model as well as the
aerosol model, including how aerosol-cloud interactions are parameterized in the cloud microphysics and an underestimation



of the aerosol absorption. Subsequent developments by Mulcahy et al. (2018) implemented in GC3.1 reduced the aerosol ERF by approximately 50% but an in-depth evaluation of the underlying aerosol properties in this model was not conducted as part of that study.

In this paper we document the GLOMAP aerosol scheme as implemented in UKESM1 and GC3.1 for CMIP6 and highlight
differences in the aerosol representation between the two models. In particular, the additional Earth system processes included in UKESM1 will lead to fundamental differences in aerosol sources, evolution and sinks that we characterize and evaluate where possible. Subsequent impacts on the aerosol forcing will be explored further in a companion paper. The present-day aerosol climatology is evaluated against observations using the coupled historical simulations conducted for CMIP6. While many process-based evaluations utilize nudged simulations, where the model's meteorology is relaxed to reanalysis data, free-
running coupled climate simulations enable feedbacks to more fully evolve due to a consistent treatment of the dynamical physical climate, biogeochemical (in the full ESM) and composition states. Evaluation of these simulations is important in establishing confidence in the predictive skill of aerosols and their feedbacks in historical and future climate simulations. This study therefore provides an assessment of the suitability of this model for wider aerosol-climate studies being conducted as part of CMIP6.

The paper is outlined as follows: Section 2 describes the host model configurations and the GLOMAP and mineral dust aerosol schemes including science updates implemented in GLOMAP since Mann et al. (2012) and Mann et al. (2010). Section 3 outlines the model simulations used in this study. Section 4 describes the observations used in the evaluation of the aerosol properties. A detailed evaluation of the tropospheric aerosol properties is presented in Section 5 followed by a discussion in Section 6.

## 2   Model description

### 2.1   HadGEM3-GC3.1 and UKESM1

In this study we evaluate the simulation of aerosols in the CMIP6 historical integrations carried out by the UK community using two global models, the physical climate model HadGEM3-GC3.1 (hereafter referred to as GC3.1) and its Earth system counterpart, UKESM1. GC3.1 is a global coupled atmosphere-ocean-ice model and details of its components and coupling
are described and evaluated at length in Kuhlbrodt et al. (2018); Williams et al. (2017). In brief, GC3.1 is comprised of the Global Atmosphere 7.1 (GA7.1) atmosphere configuration of the Unified Model (UM) (Walters et al., 2019; Mulcahy et al., 2018), NEMO ocean model (Storkey et al., 2018), CICE sea-ice model (Ridley et al., 2018) and JULES land surface model (Best et al., 2011). Here we use the low resolution version of GC3.1 (Kuhlbrodt et al., 2018) which has a horizontal resolution of approximately $135\,\mathrm{km}$ in the atmosphere and $1\,^\circ$ in the ocean. In the vertical, the atmosphere consists of 85 levels with a
model lid at $85\,\mathrm{km}$ above sea level with 50 of these levels below $18\,\mathrm{km}$ and 75 vertical levels in the ocean.

The GA7.1 model is described in detail by Walters et al. (2019) but we briefly document some of the key parameterizations of relevance for the composition and distribution of aerosols in the model. Large scale advection is modelled using the ENDGame (Even Newer Dynamics for general atmospheric modelling of the environment) dynamical core (Wood et al., 2014). ENDGame





is a non-hydrostatic, semi-implicit, semi-langrangian deep atmosphere model on a regular latitude-longitude grid. Hermite cubic vertical interpolation is used for the advection of moist prognostic variables while an improved version of the Priestley (1993) conservation scheme has been implemented for a consistent treatment of the moisture and atmospheric composition tracers. Large-scale precipitation is modelled using a single-moment scheme based on Wilson and Ballard (1999) and includes

an improved treatment of drizzle rates (Abel and Shipway (2007)). A prognostic treatment of rain allows the 3-dimensional advection of precipitation. The introduction of the latter required modifications to be made to the aerosol wet scavenging processes described in Mann et al. (2010) and is described in more detail in Section 2.6. The warm rain microphysics has undergone significant development following Boutle and Abel (2012) and Boutle et al. (2014). The Khairoutdinov and Kogan (2000) scheme for autoconversion and accretion replaces the scheme of Tripoli and Cotton (1980) that was used prior to

GA7 and has been found to significantly improve the representation of stratocumulus clouds (Boutle and Abel, 2012) and is expected to simulate more realistic aerosol-cloud-precipitation feedbacks (Boutle and Abel, 2012; Wilkinson et al., 2013; Hill et al., 2015). Large-scale cloud uses the prognostic cloud fraction and prognostic condensate (PC2) scheme (Wilson et al., 2008a, b) with modifications described in Morcrette (2012). The atmospheric boundary layer is modelled with the turbulence closure scheme of Lock et al. (2000) with modifications described in Lock (2001) and Brown et al. (2008). Convection is based

on mass flux scheme of Gregory and Rowntree (1990) with various extensions to include down-draughts (Gregory and Allen, 1991)) and convective momentum transport. The radiation scheme employed is the two-stream radiation code of Edwards and Slingo (1996) with six and nine bands in the short-wave (SW) and long-wave (LW) parts of the spectrum respectively.

The aerosol scheme employed in GA7.1 is the GLOMAP microphysical aerosol scheme (Mann et al., 2010) which is detailed in the next section. Mineral dust is simulated separately using the CLASSIC sectional dust scheme (see Section 2.7). GLOMAP

was first implemented into the HadGEM3 physical climate model configuration as part of GA7.0 (Walters et al., 2019). GA7.1 differs from GA7.0 primarily in its aerosol formulation and how aerosol-cloud interactions are parameterized. It includes an improved treatment of cloud droplet spectral dispersion, updates to the aerosol activation scheme and aerosol absorption optical properties. In addition, the seawater DMS climatology was updated to Lana et al. (2011) and the marine DMS emissions were then scaled to account for missing marine organic aerosol source. These developments, documented fully in Mulcahy et al.

(2018), reduced the excessively large, negative aerosol effective radiative forcing found in GA7.0.

The first version of UKESM1 takes GC3.1 as its physical-dynamical core and couples additional ES processes, encompassing marine and terrestrial biogeochemical cycles and fully interactive stratospheric-tropospheric trace gas chemistry (Sellar et al., 2019a). These additional ES components include the MEDUSA ocean biogeochemistry model (Yool et al., 2013), the TRIFFID dynamic vegetation model (Cox, 2001) and the stratospheric-tropospheric version of the United Kingdom Chemistry and

Aerosol (UKCA) chemistry model (Archibald et al., 2019). UKESM1, its components and details of how these different ES components are coupled are described in detail by Sellar et al. (2019a). While for the most part UKESM1 and GC3.1 models are fully traceable there are important differences related to the treatment of aerosols in these two models which we document and evaluate here. These differences primarily relate to the treatment of natural aerosol sources, aerosol chemistry and some differences in the prescription of anthropogenic $SO_2$. These differences are described in detail in the next section. While the





atmospheric timestep of the model physics is 20 minutes, due to the inherent computational cost of the chemistry and aerosol components, both components are called once per hour.

## 2.2  Aerosol scheme: GLOMAP-mode

GLOMAP is a 2-moment modal aerosol microphysics scheme simulating speciated aerosol mass and number across 5 lognor-
mal size modes. The basic aerosol model is described in detail in Mann et al. (2010) and Mann et al. (2012). Here we briefly describe the main components of the model and document in detail any updates to the previous documentation.

The configuration described and evaluated here simulates the sources, evolution and sinks of four aerosol species: sulphate ($SO_4$), black carbon (BC), organic matter (OM) and sea salt. The mineral dust scheme is described in Section 2.7. Aerosol size modes respresented are the nucleation (geometric mean dry radius, $\bar{r} < 5$ nm), Aitken ($5 < \bar{r} < 50$ nm), accummulation
($50 < \bar{r} < 500$ nm) and coarse ($\bar{r} > 500$ nm) soluble modes and an Aitken insoluble mode (see Table 1). A fixed geometric standard deviation, $\sigma_g$, for each mode is assumed. Table 1 details the properties of the aerosol size distribution used. Updates to the soluble accumulation mode width and upper size limit were introduced by Mann et al. (2012) following a detailed comparison against the bin model configuration of GLOMAP.

The aerosol composition within each mode and the mean radius of each mode is simulated according to the microphysi-
cal processes represented in the model. These include condensation of gas-phase sulphuric acid ($H_2SO_4$) and a condensible secondary organic vapour (SEC_ORG) onto pre-existing particles, aerosol coagulation within and between modes and cloud processing. Mode-merging to the next largest mode occurs when the particle geometric diameter exceeds the permitted size (see Table 1) in any given mode. New particle formation from the binary homogeneous nucleation of $H_2SO_4$ and water follows Kulmala et al. (1998) and occurs mainly in the free troposphere. Additional nucleation of new particles in the boundary layer
is not yet included. The chemical components are treated as internal mixtures within the aerosol modes. This allows for a more accurate determination of the aerosol optical properties as outlined in Section 2.8.

GLOMAP includes a prognostic treatment of sea-salt and secondary organic aerosol (SOA). Sea-salt emissions are described in Section 2.4.1. SOA is produced from the gas-phase oxidation of land-based monoterpene sources by OH, $NO_3$ and $O_3$. The molar yield of SOA from these reactions was increased from the 13% used in Mann et al. (2010) and Mann et al. (2012) to
26% in GA7 (Walters et al., 2019). This increase accounts for a wide range of uncertainty in the representation of biogenic SOA (Lee et al., 2013; Scott et al., 2014), including the large range in the observed yield of SOA; uncertainty in the emissions of precursor gases (Biogenic volatile organic compounds, BVOCs), a lack of anthropogenic and marine VOC sources as well as no contribution from isoprene in the current configuration. While BC and anthropogenic OM are emitted as insoluble species they can subsequently undergo ageing into the soluble modes by coagulation or condensation after being coated by 10
monolayers of soluble material. It is worth noting that the Aitken insoluble mode does permit particles with geometric mean dry radii greater than 50 nm. This is a result of the emitted particles from biofuel and biomass sources having emission radii of 75 nm (Stier et al., 2005). These particles will subsequently remain in this mode until they have been coated by the required 10 monolayers of soluble material. Mineral dust is simulated in UKESM1 and GC3.1 but not in the modal framework. Instead dust is simulated using the CLASSIC sectional dust scheme which is described in Section 2.4.





**Table 1.** Properties of the aerosol size distribution in GLOMAP including the permitted size range of the aerosol modes, their geometric standard deviation, $\sigma_g$, and aerosol species contributing to each mode. Species represented are sulphate, black carbon, organic matter and sea salt.

| Aerosol Mode | Geometric mean radii, $\overline{r}$ (nm) | $\sigma_g$ | Species |
|---|---|---|---|
| Nucleation sol. | $0.5 - 5$ | 1.59 | $SO_4$, OM |
| Aitken sol. | $5 - 50$ | 1.59 | $SO_4$, BC, OM |
| Accumulation sol. | $50 - 250$ | 1.40 | $SO_4$, BC, OM, SS |
| Coarse sol. | $250 - 5000$ | 2.00 | $SO_4$, BC, OM, SS |
| Aitken insol. | $5 - 50$ | 1.59 | BC, OM |

As already stated above the aerosol and chemistry routines are called hourly rather than at every dynamical model timestep. However, the aerosol emissions and boundary layer mixing of the aerosol tracers are done on every model timestep. Condensation, nucleation and coagulation processes are carried out on "competition" sub-steps to more accurately represent the competition between these processes (Mann et al., 2010; Spracklen et al., 2005). Here we employ 15 sub-steps within the
60 minute chemical timestep.

### 2.3 Anthropogenic emissions

Anthropogenic emissions of aerosols are prescribed from the CMIP6 inventories. Emissions of $SO_2$ and anthropogenic BC and OC are taken from the Community Emissions Data System (CEDS, Hoesly et al. (2018)) while biomass burning emissions are taken from van Marle et al. (2017). Biomass burning emissions of BC and OC are emitted at the surface for peat, agricultural
and savannah fires but fire emissions from forest and tropical deforestation sectors are distributed across the bottom 20 model levels (up to approximately 3 km). Biomass burning emissions are scaled by a factor of 2 following the detailed evaluation of biomass burning aerosol in Johnson et al. (2016), who found an improved agreement between observed and simulated aerosol optical depth (AOD) when this scaling factor was applied. Emissions of OC are provided in units of carbon mass, and these are scaled by a factor of 1.4 in GLOMAP (as assumed in Dentener et al. (2006)) to convert this to organic mass (OM), representing
the full mass of the organic aerosol.

$SO_2$ emissions are prescribed slightly differently in the 2 models. In GC3.1, 100% of $SO_2$ emissions from the energy sector and 50% from the industrial sector are emitted at a height of 500 m representing chimney stack emissions and associated plume rise. $SO_2$ emissions from all other sectors are emitted at the surface. In UKESM1, emissions of $SO_2$ from all sectors are emitted at the surface. This is more consistent with the treatment of the trace gas emissions (and therefore aerosol oxidants) in
UKCA which are also all emitted at the surface. As in Mann et al. (2010), we assume 2.5% of the anthropogenic $SO_2$ emissions are emitted as primary sulfate particles with an emission size distribution specified according to Stier et al. (2005).



### 2.4 Natural aerosol emissions

One of the main differences between the aerosol configurations of UKESM1 and GC3.1 lies in their treatment of natural aerosol. The fully-coupled UKESM1 model, incorporating as it does a dynamic vegetation and ocean biogeochemical model, interactively simulates emissions of marine DMS, BVOCs and primary marine organic aerosol (PMOA) (Sellar et al., 2019a).

Changes in land and ocean ecosystems have the potential to influence these natural emissions and so coupling these emissions in UKESM1 enables additional ES-aerosol climate feedbacks to be simulated. In contrast, GC3.1 either prescribes these emissions based on present-day observation-based climatologies (DMS and BVOCs) or does not include the aerosol source at all (PMOA). All natural aerosol emissions are described below:

#### 2.4.1 Sea Salt

Primary emissions of sea-salt aerosols are calculated using the bin-resolved, windspeed-dependent flux parameterization of Gong (2003). The emitted sea-salt is mapped to the accumulation and coarse soluble modes depending on whether the bin radius mid-point is below or above the upper limit of the accumulation mode size range (around 250 nm). The treatment of sea-salt emissions is the same in UKESM1 and GC3.1 apart from the specified sea-salt density which has been increased in UKESM1 from $1600 \, \mathrm{kg \, m^{-3}}$ to $2165 \, \mathrm{kg \, m^{-3}}$. The smaller density value represents a hydrated salt particle (Schulz et al., 2006)

however given GLOMAP treats the aerosol water content independently it is more accurate to use the actual dry salt density.

#### 2.4.2 DMS

In UKESM1, seawater concentrations of DMS used to drive the ocean-atmosphere flux of DMS are simulated interactively by the ocean biogeochemistry component, MEDUSA, using the parameterization of Anderson et al. (2001). As discussed in Sellar et al. (2019a) this parameterization was tuned as part of the development of the fully coupled UKESM1 model to ensure

energy balance at the top-of-atmosphere (TOA). In the Anderson et al. (2001) scheme, DMS is parameterized as a function of chlorophyll ($C$), light ($J$) and a nutrient term ($Q$) :

$$DMS = a, \qquad \log(CJQ) \leq s \qquad\qquad DMS = b[\log(CJQ) - s] + a, \qquad \log(CJQ) > s \qquad (1)$$

The fitted parameter values were originally set to be $a = 2.29$, $b = 8.24$ and $s = 1.72$. In UKESM1, the value of $a$ was tuned from 2.29 to 1.0. This essentially reduces the minimum allowed value for DMS while maintaining the slope of the fit to

observations reported in Anderson et al. (2001). DMS seawater concentrations in GC3.1 are prescribed from Lana et al. (2011). The DMS emission flux to the atmosphere is calculated in both models using the Liss and Merlivat (1986) emission scheme. In GC3.1 this emission flux is scaled by $DMS \, (1 + 0.7)$, where the additional 0.7 DMS flux represents a missing marine organic aerosol source (Mulcahy et al., 2018).





### 2.4.3 Primary marine organic aerosol

There is an increasing body of literature supporting the existence of an organic source of aerosol over the oceans from organic enriched sea-spray aerosol emitted via bubble-bursting and emission from gas-phase VOCs in the ocean surface layer (McCoy et al., 2015; O'Dowd et al., 2004; Meskhidze and Nenes, 2006; Facchini et al., 2008). Primary marine organic aerosol (PMOA)
emissions are believed to constitute the majority of the marine OA emissions (de Leeuw et al., 2011) and have been shown to have a high correlation with surface chlorophyll (Rinaldi et al., 2013; Spracklen et al., 2008). Recognising this as a potentially important source of cloud condensation nuclei (CCN) in remote marine regions, such as the Southern Ocean, GC3.1 represents this source by applying a scaling of 1.7 to the marine emissions of DMS as noted above. This is an over-simplified approach but improved the agreement of simulated and observed cloud droplet number concentrations over the Southern Ocean (Mulcahy
et al., 2018).

In UKESM1 emissions of primary marine organic aerosol are explicitly modelled following the emission parameterization of Gantt et al. (2011) with updates in Gantt et al. (2012). The organic mass fraction of the emitted sea spray aerosol, $OM_{SSA}$, is calculated as a function of the biological productivity (based on surface chlorophyll-a, $C$), the 10 m windspeed ($U_{10}$) and the sea-salt dry diameter ($D_p$) according to:

$$OM_{SSA} = \frac{\left(\frac{1}{1+\exp(X(-2.63C)+X(0.18U_{10}))}\right)}{1+0.03\exp(6.81D_p)} + \frac{0.03}{1+\exp(X(-2.63C)+X(0.018U_{10}))} \tag{2}$$

In UKESM1 we use a value of 3 for the $X$ parameter, which acts to enhance the positive correlation of $OM_{SSA}$ with $C$ and negative correlations with $U_{10}$ (Gantt et al., 2012). Chlorophyll concentrations are taken from the MEDUSA ocean biogeochemistry model with a coupling frequency of 3 hr. When used in the PMOA emission parameterization the chlorophyll concentrations are scaled by a half. This is due to systematic positive biases in the MEDUSA chlorophyll concentrations, in
particular in the Southern Ocean (Yool et al., 2013), which was found to have a detrimental impact on the emissions of PMOA. The PMOA mass emission flux is then:

$$E_{PMOA} = V_{SS} \times OM_{SSA} \times \rho_{SSA} \tag{3}$$

where $V_{SS}$ is the volume flux of emitted sea salt (in $\mathrm{cm^3\,m^{-2}\,s^{-1}}$) and $\rho_{SSA}$ is the apparent density of the emitted sea spray aerosol (in $\mathrm{g\,cm^{-3}}$) calculated as:

$$\rho_{SSA} = OM_{SSA}\rho_{OM} + (1-OM_{SSA})\rho_{salt} \tag{4}$$

where $\rho_{OM}$ and $\rho_{salt}$ are the densities of organic matter (defined as $1500\,\mathrm{kg\,m^{-3}}$) and sea salt (defined as $2165\,\mathrm{kg\,m^{-3}}$). Gantt et al. (2012) apply a global scale factor of 6 to Eqn 3 above. However, given our global PMOA emissions compare well with Gantt et al. (2012) (see Section 5.6) producing $5\,\mathrm{Tg[OM]/year}$ versus $6.2\,\mathrm{Tg[OM]/year}$ in Gantt et al. (2012) we do not





apply a global scaling factor here. This global scale factor is expected to be model dependent given the dependence of Eqn 2 on $U_{10}$ and $V_{SS}$ which will in themselves be model and resolution dependent.

In order to apply the mass emission flux calculated in Eq. 3 to GLOMAP, an assumption about the size of the emissions is required. In a mesocosm study of sea spray aerosol composition and size Prather et al. (2013) find a nascent sea spray emission

mode centred at 162 nm diameter (at 15% relative humidity). Additionally, Prather et al. (2013) find that the number fraction of this size range is dominated by primary marine organic particles, with inorganic sea salt particles dominating in the super-micron size range. These contributions are consistent with the observed mass fractions of marine aerosol in O'Dowd et al. (2004).

As implemented here, the PMOA emission calculated in Eq. 3 is distributed across the soluble (25% of mass) and insoluble

(75% of mass) Aitken modes assuming a 160 nm size. The split between soluble and insoluble is guided by the measurements in O'Dowd et al. (2004), who show insoluble marine organic aerosol concentrations a factor of three greater than soluble marine organic aerosol concentrations.

### 2.4.4  Biogenic volatile organic compounds (BVOCs)

In UKESM1 emissions of the most abundant terrestrial biogenic VOC compounds, isoprene and monoterpenes are simulated

using an interactive BVOC emission scheme, iBVOC, (Pacifico et al., 2011, 2012). In iBVOC, emissions of biogenic isoprene are based on a simplified mechanistic scheme of Pacifico et al. (2011) and the BVOC emission parameterisation from (Guenther et al., 1995) for monoterpenes. While biogenic isoprene emissions are coupled to the gas-phase chemistry in the UKCA model and thus directly affect tropospheric ozone production and methane lifetime, only emissions of monoterpenes contribute to the formation of secondary organic aerosols (SOA). As already described above the yield of SOA from monoterpene has been

doubled from 0.13 used in Mann et al. (2010) to 0.26 here.

Under present-day conditions iBVOC produces an annual global total monoterpene emission flux of approximately $115.1 \pm 1.6 \, \text{Tg[C]/yr}$ as calculated from the decadal mean of the last ten years (2005-2014) of the UKESM1 historical simulations. This is in reasonably good agreement with most state-of-art BVOC emission models (Arneth et al., 2008; Stavrakou et al., 2009; Guenther et al., 2012; Sindelarova et al., 2014; Messina et al., 2016).

In GC3.1, emissions of monoterpenes are prescribed as monthly averages. The source data is from the GEIA database which used the Guenther et al. (1995) model.

### 2.5  Aerosol chemistry

The aerosol chemistry in UKESM1 is fully coupled to the UKCA stratospheric-tropospheric (StratTrop) chemistry scheme (Archibald et al., 2019; Morgenstern et al., 2009; O'Connor et al., 2014). Therefore the chemical oxidants involved in the

oxidation of gas phase aerosol precursors, namely the hydroxyl radical (OH), ozone ($O_3$), and nitrate radical ($NO_3$), are interactively simulated and therefore have their own production and loss mechanisms. The aerosol chemical production is therefore more tightly coupled to the chemical state of the atmosphere throughout the historical simulations than in GC3.1





**Table 2.** Aerosol precursor chemistry included in UKESM1. For full details on reaction rate coefficients see Archibald et al. (2019).

| Reaction | Reference |
|---|---|
| *Gas phase reactions* | |
| $DMS + OH \rightarrow SO_2$ | Pham et al. (1995) |
| $DMS + OH \rightarrow SO_2 + MSA$ | Pham et al. (1995) |
| $DMS + NO_3 \rightarrow SO_2$ | Pham et al. (1995) |
| $DMS + O(^3P) \rightarrow SO_2$ | Sander et al. (2006); Weisenstein et al. (2006) |
| $COS + O(^3P) \rightarrow CO + SO_2$ | Sander et al. (2006); Weisenstein et al. (2006) |
| $COS + OH \rightarrow CO_2 + SO_2$ | Sander et al. (2006); Weisenstein et al. (2006) |
| $COS + hv \rightarrow CO + SO_2$ | Weisenstein et al. (2006) |
| $H_2SO_4 + hv \rightarrow SO_3 + OH$ | Weisenstein et al. (2006) |
| $SO_3 + hv \rightarrow SO_2 + O(^3P)$ | Weisenstein et al. (2006) |
| $SO_2 + OH \rightarrow SO_3 + HO_2$ | Pham et al. (1995) |
| $SO_2 + O_3 \rightarrow SO_3$ | Sander et al. (2006) |
| $SO_3 + H_2O \rightarrow H_2SO_4 + H_2O$ | Sander et al. (2006) |
| $Monoterp + OH \rightarrow 0.26 Sec\_Org$ | Atkinson et al. (1989) |
| $Monoterp + O3 \rightarrow 0.26 Sec\_Org$ | Atkinson et al. (1989) |
| $Monoterp + NO3 \rightarrow 0.26 Sec\_Org$ | Atkinson et al. (1989) |
| *Aqueous phase reactions* | |
| $HSO_3^- + H_2O_2 \rightarrow SO_4^{2-}$ | Kreidenweis et al. (2003) |
| $HSO_3^- + O_3 \rightarrow SO_4^{2-}$ | Kreidenweis et al. (2003) |
| $SO_3^{2-} + O_3 \rightarrow SO_4^{2-}$ | Kreidenweis et al. (2003) |

reflecting an additional level of realism. Changes in the concentrations of these trace gas oxidants since pre-industrial times have been shown to have important impact on the evolution of the historical aerosol forcing (Karset et al., 2018).

Table 2 describes the aerosol chemistry included in the StratTrop scheme. The reader is referred to Archibald et al. (2019) for a complete description of the StratTrop chemical mechanism in UKESM1. Gas phase emissions of $SO_2$, DMS and monoter-

5  penes are oxidised via gas phase reactions with OH, $NO_3$ and $O_3$ to eventually produce $H_2SO_4$ and SEC_ORG (see Table 2). Dissolution of $SO_2$ in cloud droplets follows the Henry's Law equilibrium approach (Warneck, 2000) and uses a global fixed value of cloud pH of 5.0. The aqueous phase oxidation rate of $SO_2$ is determined from the reaction of $HSO_3^-$ and $SO_3^{2-}$ with $H_2O_2$ and $O_3$. In the aqueous phase there is no explicit product to these oxidation reactions. Instead the reaction fluxes are used to update the accumulation and coarse mode sulphate aerosol mass. In the current configuration these calculated fluxes

10  are reduced by 25 % to account for the lack of a removal mechanism of the in-cloud $SO_4$ aerosol produced in this way.

In GC3.1 the chemical oxidants involved in the gas phase and aqueous phase oxidation of aerosol precursors are prescribed as monthly mean climatologies. As the StratTrop chemistry configuration used in UKESM1 was not finalised by the time the





GC3.1 configuration was frozen, the oxidant fields were taken from HadGEM3 simulations run for the Chemistry Climate Model Initiative (CCMI) (Hardiman et al., 2017; Morgenstern et al., 2017). The simplified "offline oxidant" chemistry scheme (see Table 3) used has only 7 atmospheric chemical tracers compared with 84 tracers used in the full StratTrop scheme and so significantly reduces the computational cost of the model. The offline oxidant scheme includes the degradation of $SO_2$ and

DMS, together with the oxidation of monoterpene to form SEC_ORG. The chemical fields $O_3$, OH, $NO_3$, $HO_2$ and $H_2O_2$ are input as time varying monthly mean fields, with only the aerosol precursor species, such as DMS, DMSO, $SO_2$, $H_2SO_4$, monoterpene and SEC_ORG retained as advected tracers. $H_2O_2$ is represented by both an advected tracer and an offline field. As it is a highly soluble species and is therefore affected by wet deposition, it is given a chemistry production and loss mechanism. The source of $H_2O_2$, $HO_2$, is not depleted and so the $H_2O_2$ concentration is not allowed to exceed the offline

field provided. The sulphur species chemical mechanism is shown in Table 3. A representation of the diurnal cycle is included by modifying the concentrations of OH and $HO_2$ using the cosine of the zenith angle and the $NO_3$ fields are reduced to zero during daylight hours.

**Table 3.** Aerosol precursor chemistry included in GC3.1.

| Reaction | Reference |
| --- | --- |
| Gas phase reactions | |
| $DMS + OH \rightarrow SO_2$ | Pham et al. (1995) |
| $DMS + OH \rightarrow 0.6 SO_2 + 0.4 DMSO$ | Pham et al. (1995) |
| $DMS + NO_3 \rightarrow SO_2$ | Pham et al. (1995) |
| $DMSO + OH \rightarrow 0.6 SO_2$ | Pham et al. (1995) |
| $SO_2 + OH \rightarrow H_2SO_4 + HO_2$ | Pham et al. (1995) |
| $Monoterp + OH \rightarrow 0.26 Sec\_Org$ | Atkinson et al. (1989) |
| $Monoterp + O3 \rightarrow 0.26 Sec\_Org$ | Atkinson et al. (1989) |
| $Monoterp + NO_3 \rightarrow 0.26 Sec\_Org$ | Atkinson et al. (1989) |
| $HO_2 + HO_2 \rightarrow H_2O_2$ | IUPAC (2001) |
| $H_2O_2 + OH \rightarrow H_2O$ | IUPAC (2001) |
| Aqueous phase reactions | |
| $HSO_3^- + H_2O_2 \rightarrow SO_4^{2-}$ | Kreidenweis et al. (2003) |
| $HSO_3^- + O_3 \rightarrow SO_4^{2-}$ | Kreidenweis et al. (2003) |
| $SO_3^{2-} + O_3 \rightarrow SO_4^{2-}$ | Kreidenweis et al. (2003) |

## 2.6   Deposition

Aerosol particles are deposited via dry and wet-deposition processes. Wet deposition includes both in- and below-cloud scav-

enging. Recent updates to the aerosol removal processes are described below.





### 2.6.1 Convective plume scavenging

Previously aerosol removal by convective precipitation was carried out after the convection scheme was called and so was removed from the levels at which the convective precipitation was formed and therefore did not interact with convective up-draught. Kipling et al. (2013) found too much aerosol aloft in the tropical upper troposphere, where the aerosol had been
transported above the heights at which aerosol removal by impaction and nucleation scavenging processes are most active. By implementing an explicit treatment of the wet scavenging of aerosol in the convective plume they found statistically significant improvements in model biases in the mass burden and vertical profiles of black carbon aerosol in remote regions. Furthermore, with the introduction of prognostic rain into the HadGEM3 model (Walters et al., 2011), the amount of large-scale precip-itation was greatly reduced improving model biases, in particular excessive light rain or drizzle events were reduced in the
model.This significantly reduced the wet deposition of aerosol and led to an increase in aerosol burden and aerosol optical depth, exacerbating the biases found by Kipling et al. (2013).

In order to avoid excessive lofting of aerosol, an in-cloud convective plume scavenging scheme is employed following the approach described by Kipling et al. (2013). The aerosol number and mass mixing ratios are depleted in the rising plume depending on the scavenging coefficients for the mode, the precipitation rate and convective updraught mass flux together with
the mass mixing ratio of liquid water and ice. The scavenging coefficients are set to be the same as those for dynamical cloud with the scavenging coefficients for accumulation and coarse modes set to 1.0. The scavenging coefficient is set to 0.5 for the soluble Aitken mode in recognition of the higher updraught velocities in convective cloud which likely lead to supersaturations high enough to activate some of the Aitken mode aerosol.The nucleation mode is not scavenged.

### 2.6.2 Nucleation scavenging

Previously the occurence of nucleation scavenging in large-scale rain was determined by the rain rate differences between a given level and the level above (Mann et al., 2010). With the implementation of prognostic rain a new approach was required. Nucleation scavenging by large-scale rain now occurs when the autoconversion and accretion rates calculated by the large-scale precipitation scheme exceed a minimum rate of $10^{-10}$ $\mathrm{kg\,kg^{-1}\,s^{-1}}$. Removal of aerosol then occurs at the rate of the conversion of cloud water to rain. All soluble mode particles with a dry radius greater than 103nm are subject to in-cloud
scavenging and insoluble mode particles are scavenged only in cold environments when temperatures are below 258 K. There is currently no representation of aerosol re-evaporation whereby aerosol particles are returned to the atmosphere when a falling cloud liquid droplet evaporates. A representation of the sub-grid variability of precipitation is incorporated by scaling the nucleation scavenging rate by the grid box mean cloud liquid fraction and assuming a raining fraction of 0.3. Removal by ice particles is included by assuming that removal by ice occurs at the same rate as the riming rate of ice crystals and aggregates
in the cloud microphysics.

Impaction scavenging of aerosol below clouds is based on Slinn (1982), using a modified Marshall-Palmer size distribution for raindrops (Sekhon and Srivastava (1971)) with raindrop terminal velocities from Easter and Hales (1983) and scavenging coefficients from Flossmann et al. (1985). Below-cloud scavenging by snow is now included following the approach described





in Wang et al. (2011) whereby a power law function is used to derive a snow scavenging rate, $k_{snow}$, for each aerosol mode, $k_{snow} = aP^b$, where $P$ is the snowfall precipitation rate. The scavenging coefficients, $a$ and $b$ for each mode are taken from Wang et al. (2011) with $b = 0.96$ for all modes and $a = 0.028$ for the nucleation, aitken and accumulation mode aerosols. Wang et al. (2011) use a value of $a = 1.57$ for coarse mode aerosols, however tests showed that this value in this model led to an

overly efficient wash-out of the larger aerosol particles, particularly at high latitudes. For a snowfall rate of $1 \, \mathrm{mm \, hr^{-1}}$ and a typical coarse mode size of $2 \, \mu\mathrm{m}$, the value of $a$ can be as low as $0.1 \, \mathrm{mm^{-1}}$ (Croft et al., 2009; Feng, 2009). A value of $a = 0.3$ is used in this configuration and is within an acceptable range of coefficients used in Feng (2009)

### 2.6.3   Dry deposition

Dry deposition and sedimentation of aerosol follows that described in Mann et al. (2010) with a modification to how the sedi-

mentation is calculated. Previously, as the aerosol deposition processes occur on the hourly timestep, the aerosol flux into and out of each model grid box was artificially restricted to ensure numerical stability. This can overly-restrict the sedimentation velocities and a sub-stepping of the sedimentation has been implemented to circumvent this problem. For computational efficiency, the sub-stepping is applied to coarse and accumulation modes only. Tests led to an optimal timestep of 15 minutes and 30 minutes being applied to the coarse and accumulation modes respectively. These changes increased coarse mode deposition

velocities, in particular impacting the sea-salt aerosol distribution. Impacts on other modes are found to be small.

### 2.7   Mineral dust: CLASSIC aerosol scheme

Mineral dust aerosol is simulated independently of the other aerosol species using the CLASSIC dust scheme (Bellouin et al., 2011). Dust aerosol can therefore be considered to be externally mixed with the GLOMAP aerosols. The CLASSIC dust scheme is used in both GC3.1 and UKESM1, though some settings differ between the models. The scheme is described in detail in

Woodward and Sellar (2019) and was developed from the scheme described by Woodward (2001) with significant updates made to the emission parameterization and the incorporation of new refractive index data for dust-radiation interactions. The dust emission parameterization was originally based on the Marticorena and Bergametti(1995) scheme. The horizontal flux is calculated in 9 bins from $0.064 \, \mu\mathrm{m}$ to $2000 \, \mu\mathrm{m}$ diameter, and from this a vertical flux in 6 bins between $0.064 \, \mu\mathrm{m}$ and $64 \, \mu\mathrm{m}$ diameter is derived. The effect of soil moisture is included using a variant of the method of Fécan et al. (1999). Dust is produced

from bare soil in both models, and also from seasonally vegetated areas of grass and shrub in GC3.1 and no preferential sources are imposed. Seasonally vegetated sources are omitted from UKESM1 in order to limit the potential impact of biases in the interactively simulated vegetation calculated by the TRIFFID scheme, which are inevitably larger than those of the IGBP climatology (Loveland and Belward, 1997) used in GC3.1. This is a relatively minor change as seasonal sources generate less than 10% of the GC3.1 dust load.

Dust is transported as six independent tracers corresponding to the emission bins and is subject to deposition through sedimentation, turbulent mixing and below-cloud scavenging. In UKESM1 the total dust deposition flux to the ocean is passed to the MEDUSA ocean biogeochemistry scheme as a source of iron for plankton growth. Unlike GLOMAP, the dust scheme is called on each 20 minute model timestep.





Three emission variables are tunable in the dust scheme: multipliers to the friction velocity and soil moisture dependence and a global dust emission multiplier. The first two are needed to compensate for the differences between the instantaneous point measurements used to derive the algorithms and the model resolved variables at the grid-scale and they also compensate for model biases. The global multiplier is a common feature of most dust emission schemes. The dust scheme was fully retuned for

UKESM1 in order to minimise biases across a number of dust metrics including near-surface dust concentrations, dust aerosol optical depth (AOD), deposition rates and size distribution against observations. This resulted in the UKESM1 emission size distribution being shifted towards larger particle sizes than in GC3.1. This affects the many size-dependent dust processes and results in changes to the global dust load, distribution and the radiative effects of dust. This evaluation is documented in detail in Woodward and Sellar (2019).

**2.8    Aerosol-radiation and aerosol-cloud interactions**

Aerosol particles can modify radiation fluxes through the direct scattering and absorption of shortwave (SW) and longwave (LW). The aerosol optical properties (refractive index, mass extinction and absorption coefficients and asymmetry parameter) of each mode are computed using the dynamically varying aerosol properties from GLOMAP. The chemical consituents of each mode are assumed to be internally mixed. The refractive index is computed as a volume-weighted average of the refractive

indices of each individual chemical component within the mode and the simulated water content. The optical properties are then determined from pre-computed look-up tables of the Mie parameter and refractive index. For mineral dust, the optical properties are calculated separately using the CLASSIC aerosol 6-bin dust scheme (Bellouin et al., 2011). In this scheme the optical properties for each size bin are fixed using pre-calculated values based on Mie calculations. These calculations assume mineral dust is hydrophobic and uses the refractive index data of Balkanski et al. (2007). The optical properties are stored in

look-up tables for use during the model integration.

Aerosol particles are activated into cloud droplets using the activation scheme of Abdul-Razzak and Ghan (2000). This scheme uses a combination of Köhler theory and empirical fits to detailed cloud parcel models to calculate the number of activated droplets from the simulated aerosol size distribution, composition, and meteorological conditions. The distribution of subgrid variability of updraught velocities is calculated according to West et al. (2014) with updates as described in Mulcahy

et al. (2018). Changes in cloud droplet number concentration ($N_d$) can impact cloud droplet effective radius (Jones et al., 2001) and also incluence the autoconversion of cloud liquid water to rain water through the Khairoutdinov and Kogan (2000) scheme, both of which affect cloud albedo (Mulcahy et al., 2018).

**3    Model simulations**

For the purpose of this evaluation we make use of the ensemble of historical simuations that were conducted with both GC3.1

and UKESM1 for CMIP6. The historical simulations cover the period from 1850 to the end of 2014 and therefore model the evolution of climate since the pre-industrial era. These simulations are forced by transient external forcings of solar variability, well-mixed greenhouse gases and other trace gas emissions and aerosols. The volcanic forcing due to the stratospheric injection



of $SO_2$ from volcanic eruptions is prescribed as a zonal mean climatology of the stratospheric aerosol optical properties over the historical period. All forcings and how they are implemented in both models are described fully in (Sellar et al., 2019b). The GC3.1 historical simulations are evaluated in full in Andrews et al. (2019). For CMIP6 a total of 19 and 4 ensemble members were run for UKESM1 and GC3.1 respectively. Each ensemble member of each model was initialised from a different date in

the respective model's pre-industrial control simulation. For the evaluation presented in this study we use the first 9 members of the UKESM1 ensemble and all 4 members of the GC3.1 ensemble. Unless otherwise stated, the ensemble mean of these models is presented in the evaluation below.

In addition we utilise the atmosphere-only configuration of each model (otherwise known as AMIP configuration) for detailed aerosol budget analysis as all of the required diagnositcs were not output in the historical simulations. Driven by observed

sea surface temperature (SST) and sea ice fields simulations were run from 1979 to the end of 2000 and allow additional simulations to be carried out at a much reduced computational cost. The UKESM1 AMIP configuration does not include the additional dynamic ocean and land surface components (Eyring et al., 2016). Instead the required vegetation (vegetation fractions, Leaf Area Index, canopy height) and surface ocean biology fields (DMS and chlorophyll) are taken from a single UKESM1 historical member and are prescribed as ancillary data, thereby maintaining traceability to the fully coupled model.

As aerosol observations for the complete historical period do not exist, we focus our evaluation on the present-day period of the historical simulations. High temporal daily or sub-daily aerosol data is not available from these simulations due to the large data volume already requested from the CMIP6 simulations. This may introduce some uncertainty into our analysis as demonstrated by Schutgens et al. (2016).

## 4 Observations

### 4.1 Surface mass concentrations

To evaluate the speciated aerosol mass concentrations we use observations from across Europe and North America, obtained from the European Monitoring and Evaluation Program (EMEP, Tørseth et al. (2012)) and the United States Interagency Monitoring of Protected Visual Environmnent (IMPROVE, Malm et al. (2004)) extensive ground-based networks. These networks provide daily observations of $SO_4$, BC and OC surface mass concentrations. Monthly mean $SO_4$ observations from East Asia

were also obtained from the Acid Deposition Monitoring Network in East Asia (EANET, https://www.eanet.asia/product/index.html). For the evaluation of $SO_4$ aerosol mass concentrations, observations were obtained from each network, where available, over the period 1980–2010. For BC and OC comparisons data was obtained over the period 1988-2014 from IMPROVE and 2003-2014 from EMEP. In order to maximise the amount of data available for comparison but also ensure the climatological representativeness of the observations we require a minimum of 8 daily mean measurements before a monthly mean is computed and

all 12 monthly means are required for an annual mean to be computed. These criteria reduce the overall number of available measurements but in total 189 (185) IMPROVE sites and 7 (4) EMEP sites provide valid measurements of BC (OC). Where measurements are provided as OC, they are multiplied by 1.4 to represent mass of OM. The observed annual means are then





compared with the simulated annual mean concentrations from UKESM1 and GC3.1 which have been linearly interpolated to the location of each station.

To evaluate the simulation of biogenic secondary organic aerosol in the models we use Aerosol Chemical Speciation Monitor (ACSM) measurements of OM from the SMEAR II station at Hyytiälä in Finland (61.85 °N, 24.28 °E, 181 m above sea level;
Hari and Kulmala (2005); Heikkinen et al. (2019)) which is surrounded by coniferous forest.

## 4.2 Aerosol optical depth

Two different types of AOD observations are used in this evaluation: ground-based measurements and satellite retrievals of AOD. Ground-based measurements are taken from the globally extensive Aerosol Robotic Network (AERONET) (Holben et al., 1998) which provides quality assured measurements of aerosol optical properties from a range of different aerosol
regimes across the globe (Holben et al., 2001). In this study, AOD at 440 nm is used from the version 2 level 2.0 product. A total of 67 stations provide valid monthly means for the period 1998–2002. From the monthly data, annual and seasonal climatological means are computed for each site and compared with equivalent model mean AOD.

Satellite retrievals of columnar aerosol properties in the visible spectrum (550 nm) provide daily, global coverage of aerosol distributions and amounts in cloud-free scenes (Kokhanovsky and de Leeuw, 2009). A number of different satellite sensors
and products are used in this evaluation. Such a satellite ensemble provides an estimate of the observational uncertainty associated with these retrievals We use monthly, area-weighted mean, column aerosol optical depth derived from visible satellite imagery. The Moderate-resolution Imaging Spectrometers (MODIS) on the Terra (MYD) and Aqua (MOD) satellites have been operating since 2000 and 2002, respectively, providing near-daily planetary coverage. Collection 6.1 of the MODIS Dark Target dataset (DT, Levy et al. (2013)) is a combination of land and ocean algorithms reporting AOD at 550 nm and a 10 km
resolution. In addition, we use the collection 6 MODIS merged data product which is a blended product of the Dark Target and Deep Blue (Hsu et al., 2004) algorithms over land and the same ocean algorithm as in MODIS-DT (Sayer et al., 2014). The Deep Blue algorithm enables retrievals over bright surfaces such as deserts and so provides additional AOD information in dust source regions. Additional sensors, the second Along-Track Scanning Radiometer (ATSR-2) operated from 1995-2003 and the Advanced ATSR (AATSR) operated from 2002-2012 and provided near-simultaneous views of nadir and 55° forward
to better constrain the surface properties in the aerosol retrieval algorithm. Three AOD products from these sensors are shown, produced as part of the European Space Agency's Climate Change Initiative (ESA CCI, Popp et al. (2016)): version 4.01 of the Optimal Retrieval of Aerosol and Cloud (ORAC, Thomas et al. (2009)), version 2.30 of ATSR Dual View (ADV, Kolmonen et al. (2016)), and version 4.3 of Swansea University's algorithm (SU, Bevan et al. (2012)). Due to the low temporal sampling resolution of the model output it is not possible to sample the model and satellite data consistently which could lead to bias in
the model-satellite comparison (Schutgens et al., 2017, 2016).

## 4.3 Aerosol number concentrations

To evaluate simulated aerosol number concentrations and hence cloud condensation nuclei we use observations of $N_{50}$ (the total particle concentration with diameter >50 nm) and $N_{tot}$ (the total particle concentration within the detectable range of the





instrument, generally 3 nm). $N_{50}$ and $N_{tot}$ observations were derived from size distribution measurements from a combination of ground-based measurements, ship-based and aircraft campaign data. The data was largely compiled as part of the Global Aerosol Synthesis and Science Project (GASSP) (Reddington et al., 2017) and is described in detail in Appendix B of Johnson et al. (2019b). The campaign data used represent campaigns in predominantly marine environments and they took place

between 1995 and 2012. All ground-based and campaign data was interpolated onto to model horizontal and vertical grid. The station data was converted to monthly means while the campaign data was assumed to be representative of the month(s) in which the campaign took place. This monthly mean data is compared with monthly mean model data averaged over the 20 year period covering 1995 to 2014 inclusive.

### 4.4   Cloud droplet number concentration

Observations of cloud droplet number concentrations ($N_d$) are limited although there has been recently more effort in this field (Grosvenor et al., 2018b). We use two satellite $N_d$ products both derived from the MODIS sensor. The first is a monthly global $N_d$ climatology at 1° resolution (Grosvenor and Wood, 2014; Grosvenor et al., 2018a, b) which retrieves $N_d$ over both land and ocean. In this dataset, retrievals of cloud optical depth and effective radius at 3.7 μm from the Level 2 MODIS Collection 5.1 cloud products are used to estimate the liquid $N_d$. See Grosvenor et al. (2018b) for details of the retrieval method and further

references. Retrievals are filtered to include only liquid cloud fractions greater than 80% and low clouds (below 3.2 km).

   The second $N_d$ monthly dataset is described in Bennartz and Rausch (2017) and retrieves information over ocean only. This dataset does not filter for high solar zenith angles and so is likely to contain overestimates at high latitudes (Grosvenor and Wood, 2014; Grosvenor et al., 2018a). It also does not filter for low-altitude clouds. Validation of $N_d$ retrievals in deeper clouds has not been performed and it is likely that some of the assumptions made for the retrieval are wrong for such clouds, although

high-altitude shallow clouds may be less problematic. In an attempt to reduce the effect of MODIS effective radius biases, this dataset is filtered to only include data points for which the effective radius from the 3.7 μm retrieval is greater than that from 2.1 μm and which is in turn greater than that from 1.6 μm, since this is the expected order based on aircraft measured droplet size profiles and the different vertical penetration depths into cloud top of the different wavelengths (e.g., see Grosvenor et al. (2018a)). However, in stratocumulus regions where $N_d$ retrievals are likely to be most reliable and where they validate well

against aircraft observations the three wavelengths produce similar effective radii values (Painemal and Zuidema, 2011). This suggests that this filtering may throw away data even in such regions and this may lead to a low $N_d$ bias (Grosvenor et al., 2018b). The time series of both products covers the period 2003 to 2014.

   Retrieved data has been removed north of 60° N and south of 60° S where retrievals are likely uncertain due to high solar zenith angles and the presence of sea-ice, which are screened for in Grosvenor et al. (2018b) but not in the Bennartz and Rausch

(2017) dataset. It is also possible that undetected sea-ice affects the Grosvenor et al. (2018b) dataset.





## 5 Results

### 5.1 Aerosol budget and burdens

Tables 4 and 5 show the global annual mean aerosol budget including the gas phase budget of aerosol precursors for UKESM1 and GC3.1. Additional spatial plots comparing the natural emissions from both models are provided in the Supplement. For
sulphate aerosol, the primary source is reflecting the 2.5% of the emitted anthropogenic $SO_2$ that is assumed to directly enter the aerosol phase, while the secondary source includes contributions from chemical production, condensation and nucleation of $H_2SO_4$. Both of these sources are in good agreement with the corresponding values reported in Mann et al. (2010). The lifetime of $SO_4$ is 5.57 days and 4.95 days in UKESM1 and GC3.1 respectively and is nearly 2 days longer than Mann et al. (2010). This appears to be due to a combination of a lower burden and higher production rates leading to a shorter lifetime in
Mann et al. (2010). The UKESM1 $SO_4$ lifetime is also on the upper end of the range shown in the AeroCom-I models (Textor et al., 2006) but compares well with a previous version of GLOMAP in HadGEM3 (Bellouin et al., 2013). The differences in lifetime across the AeroCom models and previous GLOMAP configurations reflect the diversity in aerosol processes across the different aerosol schemes but also reflect differences in the host climate models driving processes such as the aerosol tracer transport, water uptake and aerosol removal rates. For example, the frequent occurence of very low precipitation rates has been
improved considerably in recent HadGEM3 configurations (Walters et al., 2011, 2014) which has significantly reduced the aerosol nucleation and impaction scavenging rates. Indeed the convective plume scavenging (which was not included in Mann et al. (2010) or Bellouin et al. (2013)) accounts for approximately 50% of the wet scavenging in UKESM1 and GC3.1. The convective plume scavenging occurs predominantly in tropical regions and so aerosol removal rates are likely to be reduced in the mid to high latitudes compared to previous configurations. This means downwind of the key anthropogenic source regions
of Europe and North America higher concentrations of $SO_4$ aerosol are possible.

Global emissions of marine DMS are believed to be in the range of 15-35 $Tg[S]/yr$ (Lana et al., 2011) Simulated DMS emissions in UKESM1 and GC3.1 straddle this range producing 16.5 $Tg[S]/yr$ and 34.16 $Tg[S]/yr$ respectively. The much higher emissions in GC3.1 reflect the scaling applied to the DMS emissions in GC3.1 and also the different sources of DMS seawater concentrations (see Section 2.4.2). Notwithstanding the large difference in emissions the DMS burdens are compa-
rable between the two models. This implies slower oxidation of DMS to $SO_2$ in UKESM1 and subsequently a longer DMS lifetime (1.74 $days$ versus 0.85 $days$). This could be due to the different treatment of the DMS chemistry as well as differences in the availability of oxidants. For instance, the production of $SO_2$ from the oxidation by $NO_3$ is 5 times larger in GC3.1 representing approximately 50% of the DMS loss versus 25% loss in UKESM1. This offsetting of the enhanced source of natural sulfur leads to a $SO_2$ burden in GC3.1 that is only 20% higher than in UKESM1. Again differences in oxidants and smaller
wet scavenging rates in UKESM1 lead to comparable $SO_4$ burdens in the two models and a longer lifetime of 5.57 days in UKESM1 compared to 4.95 days in GC3.1. The longer $SO_4$ lifetime will likely lead to longer-range transport of $SO_4$ aerosol remote from source regions.

BC and OM emissions are in good agreement with other models (Tegen et al., 2019; Textor et al., 2006). Variations in anthropogenic sources are to be expected due to the choice of different analysis year and emission source data. The differences





in primary OM emission in UKESM1 reflect the additional source from primary marine organics which contributes of the order of $4.8\,\mathrm{Tg/yr}$. The inclusion of the iBVOC emission scheme in UKESM1 and different oxidants leads to differences in the secondary OM emission source. Again the production of SEC_ORG from the oxidation by $NO_3$ and to a lesser extent by OH is much higher in GC3.1 than in UKESM1. This suggests the availability of oxidants is very different between the two

models. In particular, the lack of a $NO_3$ sink in the offline oxidant scheme in GC3.1 is leading to a perpetual supply of $NO_3$. The lifetime of BC at 5.1 days in both models is much shorter in UKESM1 and GC3.1 than in HadGEM2-ES which suffered from an excessively long lifetime of 15 days (Bellouin et al., 2013). Overall, the BC and OM lifetimes in both models are in good agreement with each other and are approximately 1 day shorter than the AeroCom median lifetimes of 6.5 and 6.2 days respectively (Textor et al., 2006).

Mineral dust emissions initially appear high in UKESM1 compared to other models including GC3.1. However, as dust particles are emitted mainly into the larger bins these heavier particles are rapidly lost through sedimentation leading to an overall short dust lifetime of $0.86\,\mathrm{days}$ compared to $1.54\,\mathrm{days}$ in GC3.1. The different tuned settings of the CLASSIC dust scheme in UKESM1 and GC3.1 lead to global dust emissions which are approximately double in the former. It should be noted that due to the structure of the dust code, the dust emission diagnostics include all particles released from the surface, including

large particles which are almost immediately re-deposited within a single timestep and so are not transported or interacting with the model in any way. These very short-lived dust particles are also included in the diagnosed deposition values and therefore lifetime. This hampers quantitative comparison with other models and observations. Nevertheless, the global dust burden of both models only differs by approximately $4\,\mathrm{Tg/yr}$ (25%) and compares well with other models (Tegen et al., 2019; Textor et al., 2006). These factors are discussed and evaluated in more detail in Woodward and Sellar (2019).

There is a very large uncertainty in global sea-salt emissions (Lewis and Schwartz, 2004) with the UKESM1 and GC3.1 global means well within this range. Both emission and lifetime are in good agreement with the AeroCom median value of 6280 Tg/yr and 0.41 days respectively (Textor et al., 2006). The global mean sea-salt emissions and burdens are higher in UKESM1 by approximately 35% due to the change in the prescribed sea-salt density as well as differences in the 10m windspeeds.

The spatial distributions of the annual mean gas-phase $SO_2$ and aerosol burdens from the historican ensemble mean are broadly similar in both models (Figures 1 and 2) with a few noteworthy differences. The higher $SO_2$ burden in GC3.1 (Figure 1a) is globally widespread with particularly notable enhancements in the tropical regions. In this region the Lana et al. (2011) seawater DMS concentrations in GC3.1 peak and will be significantly higher than the seawater concentrations calculated interactively in UKESM1. As discussed above $SO_4$ burdens are globally comparable between the two models. Lower burdens

are found in the Southern Hemisphere high latitudes in GC3.1 (Figure 1b). This is likely caused by the longer lifetime in UKESM1 leading to long-range transport of $SO_4$ to remote high latitudes.

BC burdens compare extremely well between both models (Figure 2c and d). Differences in OM burdens (Figure 2e and f) are small and are found primarily over North America and Canada where the burden is higher in UKESM1 while a lower burden is found in tropical biomass burning regions. Sea-salt burdens (Figure 2g and h) are higher in UKESM1 over all ocean basins.

UKESM1 has a higher dust burden (Figure 2i) across the Northern Hemisphere due to the higher dust emissions as discussed



**Table 4.** Aerosol and gas phase sulphur budget for UKESM1. Units for production and loss fluxes are in Tg[species]/yr except for sulphate aerosol, $SO_2$ and DMS which aerosol reported as Tg[S]/yr. The value in the parenthesis in "Wet" scavenging column is the % that is scavenged via convective plume scavenging. There is no plume scavenging for $SO_2$ or mineral dust. The values are calculated from an 18 year AMIP simulation covering the period 1980-1998.

| Species | Production (Tg/yr) | | Loss (Tg/yr) | | | Burden (Tg) | Lifetime (days) |
|---|---|---|---|---|---|---|---|
| | Primary | Secondary | Dry | Wet | Oxidation | | |
| Sulphate | 1.87 | 49.59 | 7.14 | 36.06 (49%) | | 0.67 | 5.57 |
| $SO_2$ | 74.65 | 16.71 | 28.91 | 13.40 (N/A) | 49.59 | 0.53 | 2.08 |
| DMS | 16.48 | – | – | – | 16.57 | 0.08 | 1.74 |
| BC | 9.05 | – | 2.68 | 6.36 (56%) | – | 0.13 | 5.13 |
| OM | 66.51 | 35.58 | 22.61 | 79.25 (61.2%) | – | 1.5 | 5.31 |
| Sea-Salt | 5490.0 | – | 3415.73 | 2076.78 (33%) | – | 7.34 | 0.48 |
| Dust | 7398.55 | – | 6453.62 | 936.0 (N/A) | – | 17.71 | 0.86 |

**Table 5.** Aerosol and gas phase sulphur budget for GC3.1. Units for production and loss fluxes are in Tg[species]/yr except for sulphate aerosol, $SO_2$ and DMS which aerosol reported as Tg[S]/yr. The value in the parenthesis in "Wet" scavenging column is the % that is scavenged via convective plume scavenging. There is no plume scavenging for $SO_2$ or mineral dust. The values are calculated from an 18 year AMIP simulation covering the period 1980-1998.

| Species | Production (Tg/yr) | | Loss (Tg/yr) | | | Burden (Tg) | Lifetime (days) |
|---|---|---|---|---|---|---|---|
| | Primary | Secondary | Dry | Wet | Oxidation | | |
| Sulphate | 1.87 | 57.02 | 6.91 | 42.77 (47%) | | 0.68 | 4.95 |
| $SO_2$ | 74.64 | 33.01 | 30.46 | 20.19 (N/A) | 57.02 | 0.68 | 2.8 |
| DMS | 34.0 | – | – | – | 33.02 | 0.08 | 0.85 |
| BC | 9.05 | – | 2.58 | 6.48 (56%) | – | 0.13 | 5.15 |
| OM | 61.64 | 40.15 | 21.55 | 80.34 (62%) | – | 1.45 | 5.11 |
| Sea-Salt | 4077.74 | – | 2301.17 | 1778.11 (34.5%) | – | 6.81 | 0.60 |
| Dust | 3102.14 | – | 2572.33 | 525.93 (N/A) | – | 13.22 | 1.54 |

above. The dust burden in also higher over Australia but lower in South America and South Africa. The lower burdens in these regions are due to lower emissions from the Atacama and Kalahari deserts and reflects the different vegetation properties of the two models.





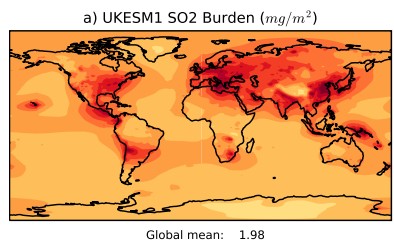
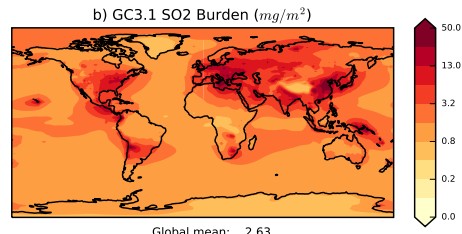

**Figure 1.** Annual mean $SO_2$ burden (mg $[SO_2]$ m$^2$) in (a) UKESM1 and (b) GC3.1. The annual mean burdens are computed from the 9 and 4 member historical ensemble means from UKESM1 and GC3.1 respectively from years 1980 to 2014 inclusive.

## 5.2 Aerosol mass concentrations

### 5.2.1 Sulphate aerosol

Figure 3 compares simulated annual mean $SO_4$ concentrations from both UKESM1 and GC3.1 with observations from the EMEP, IMPROVE and EANET networks. The IMPROVE sites have been split into east and west IMPROVE to help distin-
guish between the larger number of $SO_2$ source regions historically found across eastern North America and the cleaner west coast. The location of all sites from all networks is also shown in Figure 4. Overall, both models compare relatively well with observations from all 3 networks with simulated concentrations being generally within a factor of 2 of those observed. Comparison over Europe (Figure 3a and b) show a high degree of scatter but overall both models tend to be negatively biased. UKESM1 underestimates the observations to a greater extent than GC3.1 (larger normalised mean bias, NMB) and has a
slightly larger root mean square error (RMSE).

Over North America, both models systematically underestimate the observations in the east of the country but show a high correlation ($r^2 > 0.8$). In the west, the models generally tend to overestimate the observations and have a lower correlation ($0.2 < r^2 < 0.4$) due to the larger degree of scatter in this region. This region is expected to be relatively remote from emission sources which are predominantly in the east and so the positive bias highlights potential issues in the amount of sulphate or
$SO_2$ transported from source, excessive oxidation away from source or too low removal rates.

The simulated $SO_4$ is underpredicted by both models across East Asia in the EANET measurement locations (Figures 3c and d). Overall, the simulated $SO_4$ in UKESM1 tends to show a larger underestimation of observations than GC3.1 with a normalised mean bias (NMB) of -0.21 compared to -0.18 in GC3.1. This is likely due to the differences in $SO_2$ emissions and oxidants between the models.

Figure 4 shows a time series of simulated and observed annual mean $SO_4$ concentrations averaged across all of the available measurement locations in each network and for all available years within a particular region. The observed reduction in $SO_4$ concentrations across Europe (EMEP) is shown in Figure 4a and is reproduced well by both GC3.1 and UKESM1, although there is a consistently larger underprediction in UKESM1. for all years. However both models sit within the observed variability. This underprediction of the annual mean surface $SO_4$ concentrations over Europe is in agreement with Turnock
et al. (2015) who also showed this underestimation was dominated by a low bias in wintertime while the summertime surface

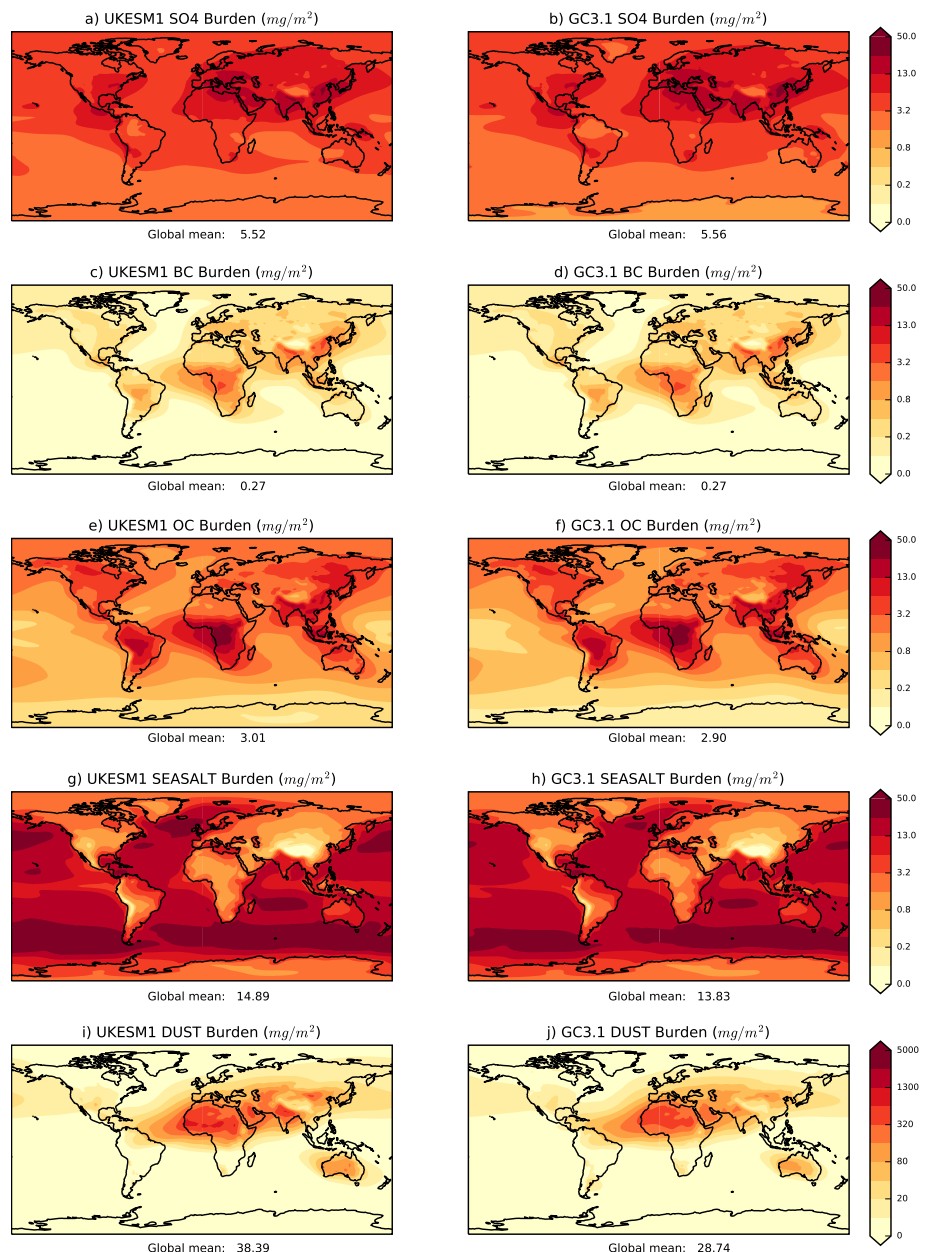

**Figure 2.** Annual mean burden $(\mathrm{mg\,m^{-2}})$ of (a,b) $SO_4$, (c,d) BC, (e,f) OC, (g,h) sea-salt and (i,j) mineral dust aerosol in (left column) UKESM1 and (right column) GC3.1. The annual mean burdens are computed from the 9 and 4 member historical ensemble means from UKESM1 and GC3.1 respectively from years 1980 to 2014 inclusive. Note the different contour levels in panels i and j.

$SO_4$ was overestimated in a previous HadGEM3-UKCA configuration. Examining the seasonal cycle in UKESM1, we find



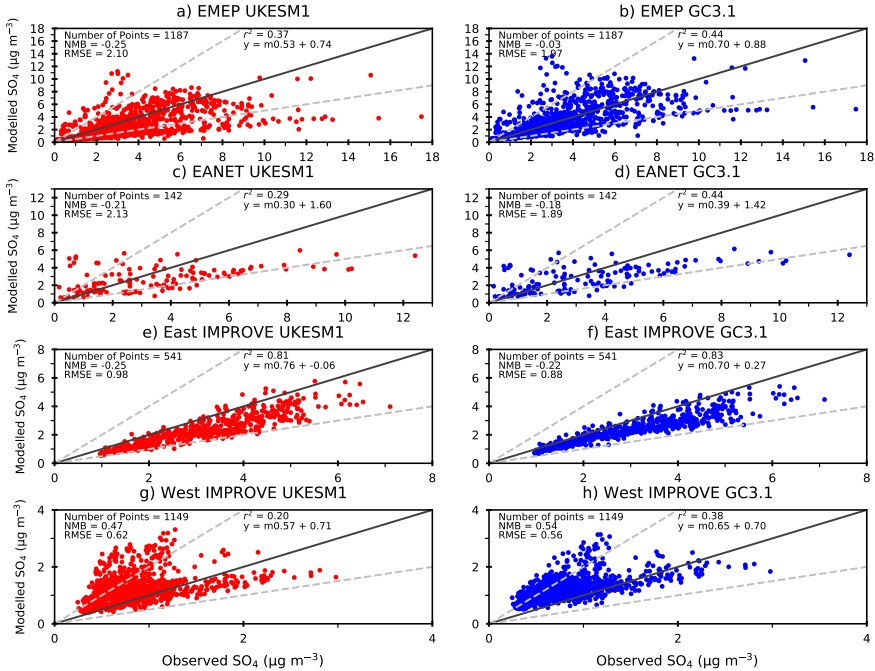

**Figure 3.** Comparison of simulated annual mean $SO_4$ from (left column) UKESM1 and (right column) GC3.1 against ground-based measurements from (a,b) EMEP (Europe), (c,d) IMPROVE (North America) and (e,f) EANET (East Asia) networks. Observations and model data cover the years 1980 to 2010 for EMEP, 1988 to 2010 for IMPROVE and 2000 to 2010 for EANET. Normalise mean bias (NMB), root mean square error (RMSE), correlation coefficient ($r^2$) and linear regression statistics are also included. Distribution of network stations is shown in Figure 4.

that the model does underestimate wintertime $SO_4$ while summertime concentrations are in much better agreement with the observations (not shown).

Over North America, a small negative trend in $SO_4$ concentrations is found in the East IMPROVE sites and is generally well represented by both the models although again a negative model bias is found. The absolute $SO_4$ concentrations and

5 negative trend is generally smaller at these sites than over Europe although the observations cover a shorter time period. Over the western measurement sites there is little to no trend found in the observation with the models exhibiting a small negative trend and generally overestimating the observations. Finally, over East Asia a general increase in the $SO_4$ concentrations is found reflecting the increase in anthropogenic emissions in this region over the observed period, particularly in China. Again the models are able to capture the rising trend although concentrations are consistently underpredicted in both models for all

10 years.



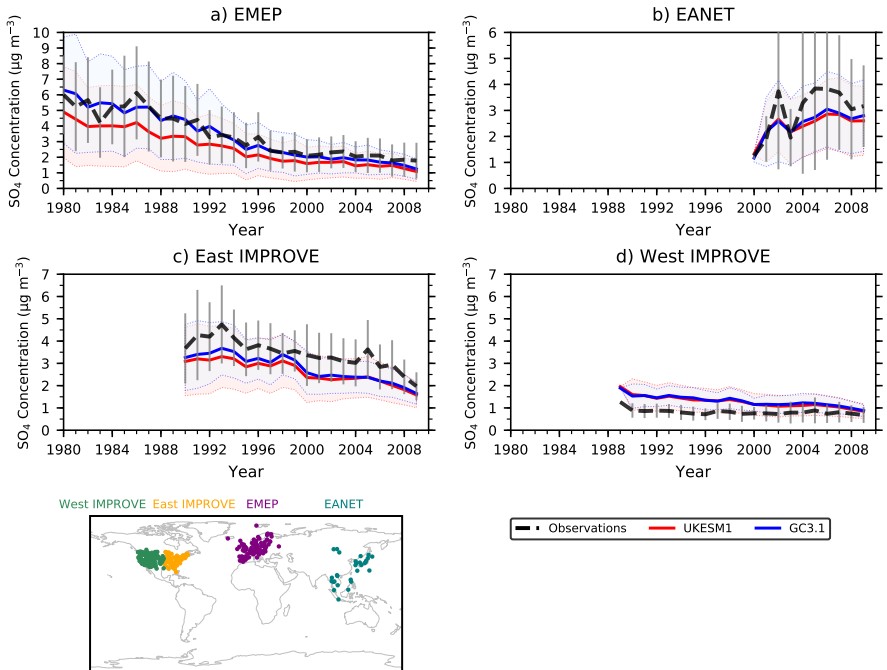

**Figure 4.** Time series of annual mean observed (dashed lines) and simulated (solid lines) sulphate concentrations averaged across all measurement locations in each network for a particular year. Error bars and shaded areas show ±1 standard deviation of the observed and modelled annual mean values across all the measurement locations.

### 5.2.2 Carbonaceous aerosol

Figure 5 compares the annual mean simulated BC and OM mass concentrations with ground-based observations from both IMPROVE and EMEP networks. The evaluation is heavily weighted to the IMPROVE measurements due to the much larger number of observations available from this network both in terms of number of sites and observation period (See Section 4). Simulated BC (Figure 5a and b) from both UKESM1 and GC3.1 agree reasonably well with the measurements with a correlation coefficient of 0.4. Both models have a small negative bias with GC3.1 exhibiting a slightly larger bias than UKESM1. The broadly similar performance of both models is not surprising considering the consistent treatment of BC emissions in both models. The relatively low correlation and high degree of scatter is not totally unexpected given the model simulations are free-running with no relaxation towards observed meteorological conditions. Accurate temporal and spatial sampling is known to be important in model-observation comparisons and evaluation of annual data here is likely to introduce some uncertainties in our comparison (Schutgens et al., 2017). However given the relatively strict criteria applied to the observed data in building the annual mean observed climatology we believe the observed data are representative of the annual climatology at each site.

Figure 5c and d compares simulated annual mean OM mass concentrations from UKESM1 and GC3.1 respectively against ground-based measurements from IMPROVE and EMEP. UKESM1 compares well with the observations although the root



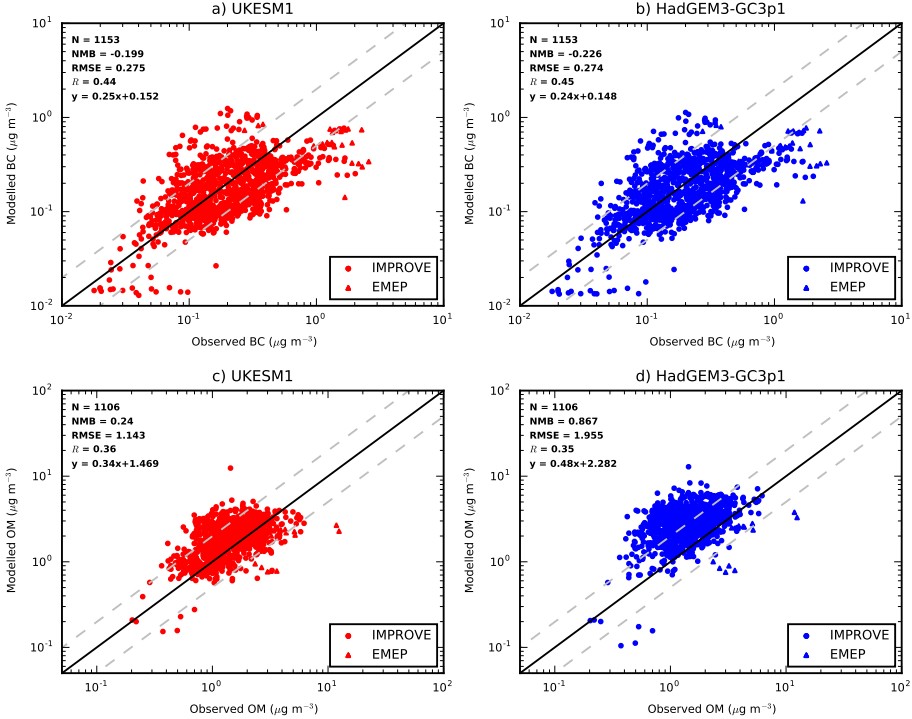

**Figure 5.** Comparison of simulated annual mean BC and OM against ground-based measurements from the IMPROVE and EMEP netorks for (a,c) UKESM1 and (b,d) GC3.1. Observations and model data cover the years 1988 to 2015. Normalised mean bias (NMB), root mean square error (RMSE) and linear regression statistics are also included.

mean square error (RMSE) is higher than for BC. Both models show a reasonable correlation with the measurements and tend to be positively biased. GC3.1 has much stronger positive bias than UKESM1 with a normalized mean bias (NMB) that is 3 times larger than UKESM1 (Figure 5d). Given the strong weighting of this comparison to measurement sites across North America the different behaviour in the models for surface OM mass concentrations is likely due to lower contributions to the

5  OM mass from SOA in UKESM1 in this region. Emissions of monoterpenes over North America from the iBVOC model in UKESM1 will be different to the prescribed emissions in GC3.1 (see Figure 2 in the Supplement). It should be noted that the prescribed emissions used in GC3.1 are also model based and may suffer from biases in simulated vegetation fractions and types. The oxidation of emitted monoterpene to SEC_ORG is also different in these two models. The global production of SEC_ORG in GC3.1 is over 50% larger than in UKESM1, in particular oxidation by $NO_3$ is more than doubled highlighting

10  a possible limitation of the offline oxidant chemistry in GC3.1 where there is no sink for this species. The inclusion of an interactive emission source which more accurately reflects the change in vegetation distribution and response to changing temperatures combined with interactive oxidants in UKESM1 leads to an improved simulation of OM in this region.

To further evaluate the different treatment of biogenic sources in the physical and ES models and its influence on the evolution of SOA we now examine the seasonal cycle of organic aerosol at a remote forested site, Hyytiälä in Finland (Figure





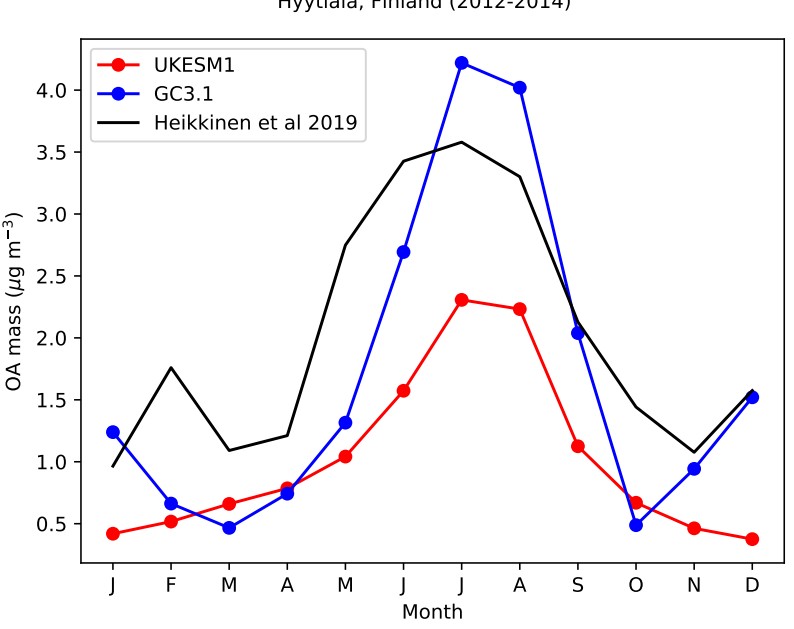

**Figure 6.** Comparison of the simulated monthly means of organic aerosol mass concentrations from UKESM1 and GC3.1 against measurements made at Hyytiälä, Finland. The comparison covers the years 2012-2014 inclusive.

6). As monoterpene emissions from vegetation are highly temperature dependent a strong seasonal cycle in OM is observed (see also Figure 2 of the Supplement). At Hyytiälä biogenic sources contribute to the peak in observed OM between May and August. Both models simulate a seasonal cycle with GC3.1 generally simulating higher concentrations of OM than UKESM1. GC3.1 shows a reasonable agreement with the observations but concentrations peak later by about 1 month and the model
generally underestimates concentrations in other months. UKESM1 has a weaker seasonal cycle than both GC3.1 and the observations and underestimates the observations at Hyytiälä in all months. The prescribed emissions of monoterpene in this region are higher in GC3.1 than in UKESM1 in both winter and summer months (see Figure 2 of the Supplement) explaining the differences in simulated OM mass concentrations over this forested site.

### 5.3 Aerosol optical depth

To evaluate the evolution and global distribution of aerosol optical depth in UKESM1 and GC3.1 we use a combination of ground-based and satellite retrievals of AOD. The timeseries of global annual mean AOD at 550 nm from the UKESM1 and GC3.1 ensembles is plotted from 1979 to 2014 in Figure 7. The global mean AOD in GC3.1 is consistently higher than UKESM1 in all years by approximately 0.01 but both models show the same inter-annual variability driven primarily by changes in emissions. The individual historical members of each model ensemble are also plotted as is the UKESM1-AMIP
simulation. The variability in global mean AOD among the individual members is small for both models and the AMIP simu-





lation is also in good agreement with the UKESM1 ensemble mean demonstrating good traceability from the fully coupled ES to the atmosphere-only configurations.

Also plotted in Figure 7 are the annual mean AOD from a number of satellite retrieval products using the MODIS, ATSR and AATSR-2 sensors (see Section 4.2) The satellite retrievals are plotted for the retrieval period of each satellite sensor. While a

comparison of the different satellite products is beyond the scope of this work it is clear there is a significant uncertainty in the retrieved AOD from the different satellite sensors and aerosol retrieval algorithms. There are numerous sources of uncertainty in satellite remote sensing datasets that can explain the differences shown (Povey and Grainger, 2015). The MODIS and ATSR instruments have different swath widths and overpass times, such that they can observe significantly different aerosol regimes. Biases due to surface albedo, differences in the cloud clearing algorithms and the assumed aerosol microphysical properties

are also likely (Popp et al., 2016). Multiple retrievals are used here to provide an indication of the observational uncertainty in AOD. Indeed the inter- and intra-model spread in AOD is much smaller than the spread in satellite retrieved AOD which spans a range of appoximately 0.04. During the period of satellite observations (from 1995 onwards), UKESM1 and GC3.1 are within the range of the satellite AOD although UKESM1 generally sits at the lower end of the observed range. Both models are in good agreement with the MODIS-DT (MYD/MOD), Swansea (SU) and ADV global mean AOD products. Higher AOD

in the merged MODIS (MODIS c6) product is expected due to the addition of the Deep Blue retrieval which will include AOD retrievals over dust source regions. But as is discussed below differences in the spatial distribution of the satellite AOD can result in misleading interpretation of global mean values. We now compare the seasonal spatial distributions of a subset of the satellite retrievals shown in Figure 7 for winter (December-January-February, DJF) and summer (June-July-August, JJA).

The seasonal spatial distribution of AOD, shown in Figure 8, highlights some notable regional differences in AOD between

UKESM1 and GC3.1 (Figures 8a-d). Reflecting the annual AOD differences already noted above, UKESM1 has lower global mean AOD than GC3.1 in both DJF and JJA. Across high latitude ocean basins UKESM1 has higher AOD in the respective winter seasons of each hemisphere. This reflects the higher sea-salt emission in UKESM1 which will peak in the winter months. In JJA, GC3.1 has higher AOD across most of the northern hemisphere. This is driven in part by the elevated DMS emissions in the Northern Hemisphere but also the perpetual supply of oxidants in GC3.1 leads to higher production of $SO_4$

aerosol downwind of source regions. For instance, a higher optical depth is clearly seen over the Mediterranean downwind of the volcanic $SO_2$ source from Mount Etna in Sicily. Dust AOD is clearly lower in UKESM1 in both seasons with lower AOD evident over dust sources and in the dust outflow region from the Sahara desert. Despite higher emissions and burden in UKESM1 the difference in the dust size distribution between the two models, as highlighted above, results in larger sized particles which are less optically efficient at the mid-visible wavelengths studied here. Over South America, differences in

monoterpene emissions plus differences in the oxidising capacity of atmosphere leads to lower AOD in UKESM1.

The spatial distribution of AOD among the satellite AOD products also show interesting regional features (Figure 8e-j). ORAC AOD tends to be higher than both MODIS and SU AOD in both seasons but has lower AOD over dust source regions. SU retrieves much lower AOD over ocean regions but has higher dust AOD than other products picking up dust sources over Australia and South America not captured by the other satellite datasets or models. In comparing the simulated and

observed AOD distributions it is important to note that the models and satellite datasets have not been consistently temporally





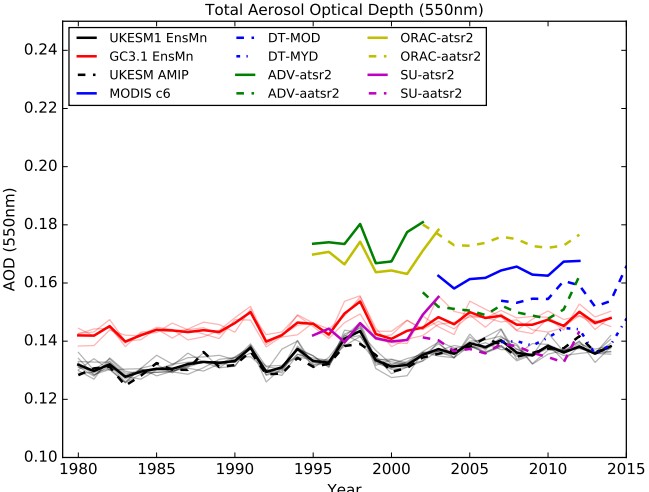

**Figure 7.** Timeseries of simulated annual mean aerosol optical depth (AOD) at 550nm from the UKESM1 and GC3.1 historical ensemble means, UKESM1-AMIP with multiple satellite AOD products. Also plotted are the individual ensemble members from each model (light shading) and the UKESM1-AMIP simulation (dashed black line). The satellite products are the collection 6 MODIS merged dataset (MODIS c6), MODIS Dark Target from Terra (DT-MYD) and Aqua (DT-MOD) satellite sensors, the ATSR Dual-View dataset (ADV), the Optimal Retrieval of Aerosol and Cloud dataset (ORAC) and the Swansea University dataset (SU).

or spatially sampled due to the low temporal resolution AOD output of the CMIP6 historical simulations. This will likely lead to uncertainties in the resulting evaluation (Schutgens et al., 2017, 2016) so the comparison presented here is qualitative but provides a reasonable indication of the representativeness of the simulated AOD climatology and provides useful insight into the gross, systematic biases in the model.

Notwithstanding these observational differences and associated uncertainties the spatial distribution and seasonal cycle of simulated AOD agrees well with the satellite observations, capturing the elevated AOD in JJA and lower AOD in DJF (Figure 8). In JJA the models broadly capture the elevated AOD over the Northern Hemisphere continents including the contrasting AOD signal over North America with lower AOD across the western United States and higher AOD on the East coast. Boreal forest fires, the primary contributor to summertime AOD across Canada, Alaska and Russia, are accurately captured and AOD from

biomass burning sources in the tropics is within the observational range. As already mentioned, both models underestimate the dust AOD over and downwind of key source regions. This leads to low biases in AOD over the Sahara and across the tropical Atlantic Ocean and Arabian Sea. Across the Southern Ocean, AOD appears in reasonable agreement with the observations but both models tend to underestimate in DJF and overestimate in JJA. This is in agreement with the recent evaluation of Southern Ocean aerosol in the atmosphere only configuration of GC3.1 conducted by Revell et al. (2019). The evaluation over

high latitude regions should be treated with particular caution due to the relatively low number of satellite retrievals at these latitudes in winter.





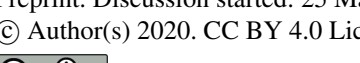

**Figure 8.** Aerosol optical depth (AOD) at 550nm from (a,b) UKESM1 and (c,d) GC3.1 historical ensemble and multiple satellite products for (left) DJF and (right) JJA. The satellite products are (e,f) MODIS merged dataset (collection 6), (g,h) ORAC, (i,j) Swansea products as fully described in the text.



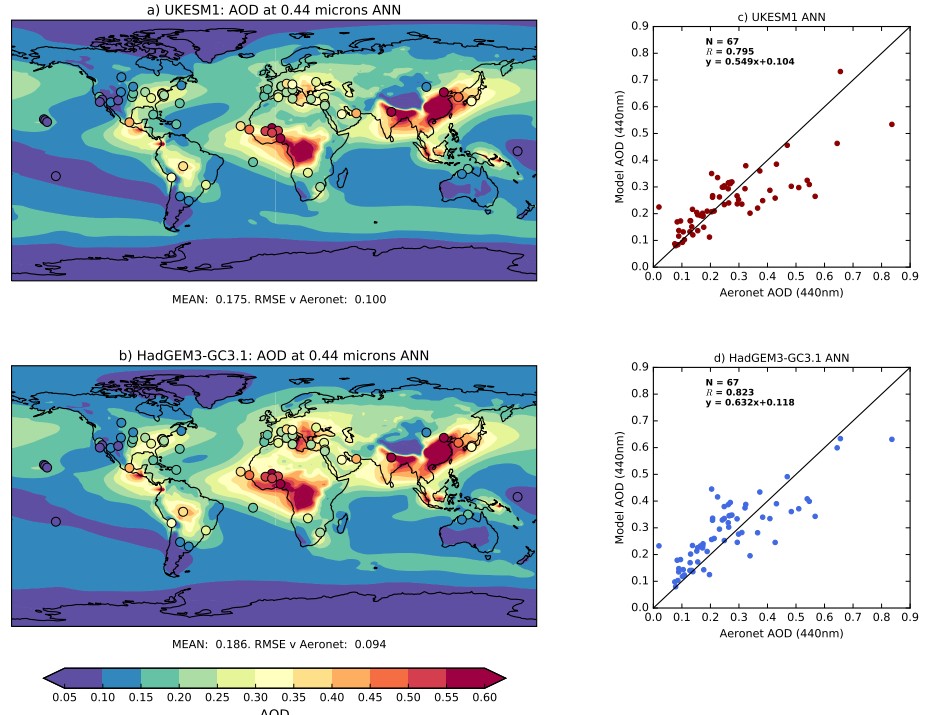

**Figure 9.** Annual mean aerosol optical depth at 440nm from (a) UKESM1 and (c) GC3.1. Ground-based aerosol optical depth retrievals at various Aeronet sites are overlaid in circles using the same colorscale. Scatterplot of simulated and observed AOD values at Aeronet sites for (b) UKESM1 and (d) GC3.1.

The model disparity in AOD over the North Atlantic and Pacific Oceans in JJA is difficult to assess given the relatively low AOD in the SU dataset and much higher AOD in MODIS and ORAC datasets. UKESM1 underestimates the MODIS and ORAC AOD in this region but agrees well with SU AOD and the opposite biases are found in GC3.1. Disparity amongst the observations makes evaluation of AOD over remote oceans difficult.

5    Figure 9 compares the annual mean AOD from both models with ground-based AERONET observations. AERONET sun-photometers provide a direct measurement of the attenuation of sunlight due to aerosol and so are not affected by the same large uncertainties as are satellite retrievals of AOD, for instance uncertainty in the underlying surface properties. The high measurement frequency from these long-term observing sites provides us with a globally representative climatology at 67 sites shown in Figure 9a and b. Although notably we are missing sites at high latitude locations and over remote oceans. Spatially

10   both models show excellent agreement with the observations and Asian, European and American AOD are all well captured. While both models show a high degree of correlation with the observations (Figure 9c and d) (correlation coefficient, R > 0.79) GC3.1 shows a slightly higher correlation due to the smaller bias in the dust-prone sites in North Africa.





## 5.4 Aerosol number

One of the key advantages of 2-moment aerosol schemes such as GLOMAP over simpler bulk schemes is the ability to interactively simulate the evolution of the aerosol number size distribution. Here we assess the skill of the simulated aerosol number concentrations at size ranges important for influencing cloud droplet formation. There is some uncertainty in this evaluation

given the observations largely represent campaign data from short (often a single month) periods that often target specific aerosol regimes. While the model and observations have been sampled consistently from the same month and altitude, the extracted $N_{50}$ and $N_{tot}$ concentrations have then been averaged in the vertical and over all months to illustrate the overall annual mean bias at each observation location. This provides a more representative view of the coupled model's ability to simulate the mean climatatology (Watson-Parris et al., 2019). The simulated $N_{tot}$ and $N_{50}$ is the integrated number concentration of all

particles with a diameter greater than 3 nm and 50 nm respectively.

Overall, higher number concentrations are found in UKESM1 compared to GC3.1 across all size modes (not shown). This could be due to the different treatment of natural emissions, notably DMS and the inclusion of PMOA, but also could reflect an increase in the binary homogeneous nucleation rate when coupled to the interactive chemistry model. Figure 10 plots the concentrations of $N_{50}$ from UKESM1 and GC3.1 at the locations of the observations. The ratio of model to observed values

(Figure 10d and e) demonstrates that both models are generally within a factor of 2 of the observations. In general, UKESM1 has higher $N_{50}$ than GC3.1 globally which acts to reduce negative biases over the Northern Hemisphere continents and high latitude oceans but increases a positive bias in other ocean region basins. $N_{50}$ in the stratocumulus region off the west coast of North America also has a positive bias in both models and appears to be slightly more enhanced in UKESM1. While there is a notable absence of observations in the Southern Hemisphere high latitudes, data from ACE1 (Clarke et al., 1998) off the

Southeast coast of Australia shows an underestimation of $N_{50}$ in GC3.1. The introduction of the PMOA source in UKESM1 clearly increases the $N_{50}$ in this region (by approximately $50 \, cm^{-3}$ in the austral summer) and generally improves the bias although a positive bias is introduced at some grid points.

Similar to $N_{50}$, simulated $N_{tot}$ (Figure 11) in both models is generally within a factor of 2 of the observations. Over most ocean regions, with the exception of high latitudes, both models tend to overestimate $N_{tot}$ with the positive bias slightly worse

in UKESM1. An underestimation is generally found over Northern Hemisphere continents in both models with GC3.1 also underestimating $N_{tot}$ downwind of anthropogenic source regions off the east coast of North America. In high latitude regions, such as the Southern and Arctic oceans, both models tend to underestimate $N_{tot}$ with a larger negative bias found in GC3.1. In the Southern Ocean, the tendency to overestimate $N_{50}$ and underestimate $N_{tot}$ could reflect inaccuracies in the prescribed emission size distribution of the PMOA which potentially leads to too much of the PMOA residing in the accumulation mode.

It could also point to missing sources such as the absence of a boundary layer nucleation scheme in the models.

Overall, the differences in the $N_{50}$ and $N_{tot}$ model biases are small although the model to observed ratio is higher for $N_{50}$ over the remote oceans than for $N_{tot}$ highlighting potential biases in $N_{50}$ and subsequently CCN. However, definitive conclusions on the model performance here are difficult to draw, given the potentially large inter-annual variability in these simulated variables combined with the uncertainty in the observations (Johnson et al., 2019a; Watson-Parris et al., 2019).



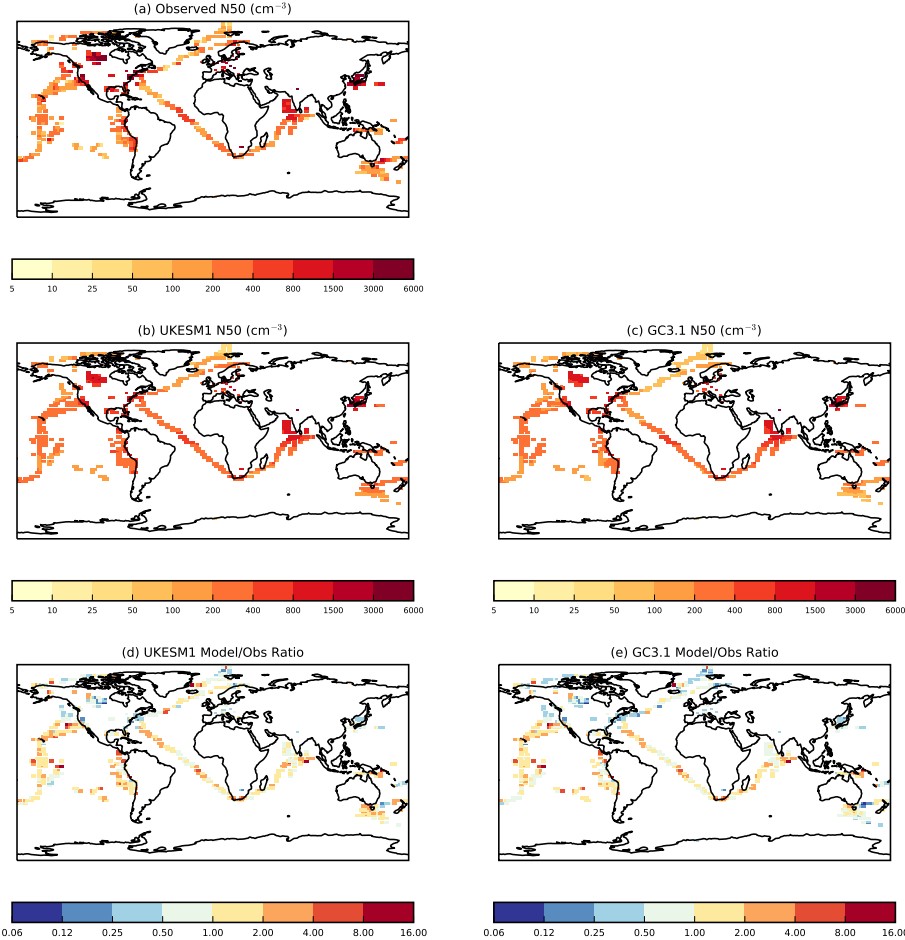

**Figure 10.** Comparison of simulated $N_{50}$ (total particle concentration with diameter > 50 nm) (cm$^{-3}$) from (a) UKESM1 and (b) GC3.1 against (c) gridded observations from a combinations of ground-based, ship and aircraft campaigns. The ratio of model to observed data for (d) UKESM1 and (e) GC3.1.

## 5.5 Cloud droplet number concentration ($N_d$)

The annual mean spatial distribution of $N_d$ from the two satellite $N_d$ products, UKESM1 and GC3.1 is shown in Figure 12. It is first important to note the large difference in $N_d$ between the two satellite products, with the Grosvenor $N_d$ being systematically higher than the Bennartz $N_d$ (on average 30% higher over ocean, Figure 12a and b). This is likely due to the different filtering techniques applied based on the effective radii retrieved at different wavelengths in the Bennartz dataset, which is likely to lead to an underestimate in $N_d$ as discussed in Section 4.4. In addition, the restriction to cloud fractions greater than 80% in the Grosvenor dataset, whilst likely giving more accurate retrievals, may lead to overestimates compared to datasets where this sampling is not performed due to the positive correlation between cloud fraction and $N_d$. This highlights the inherent difficulties in retrieving $N_d$ from space, further complicating our ability to constrain this variable in models.





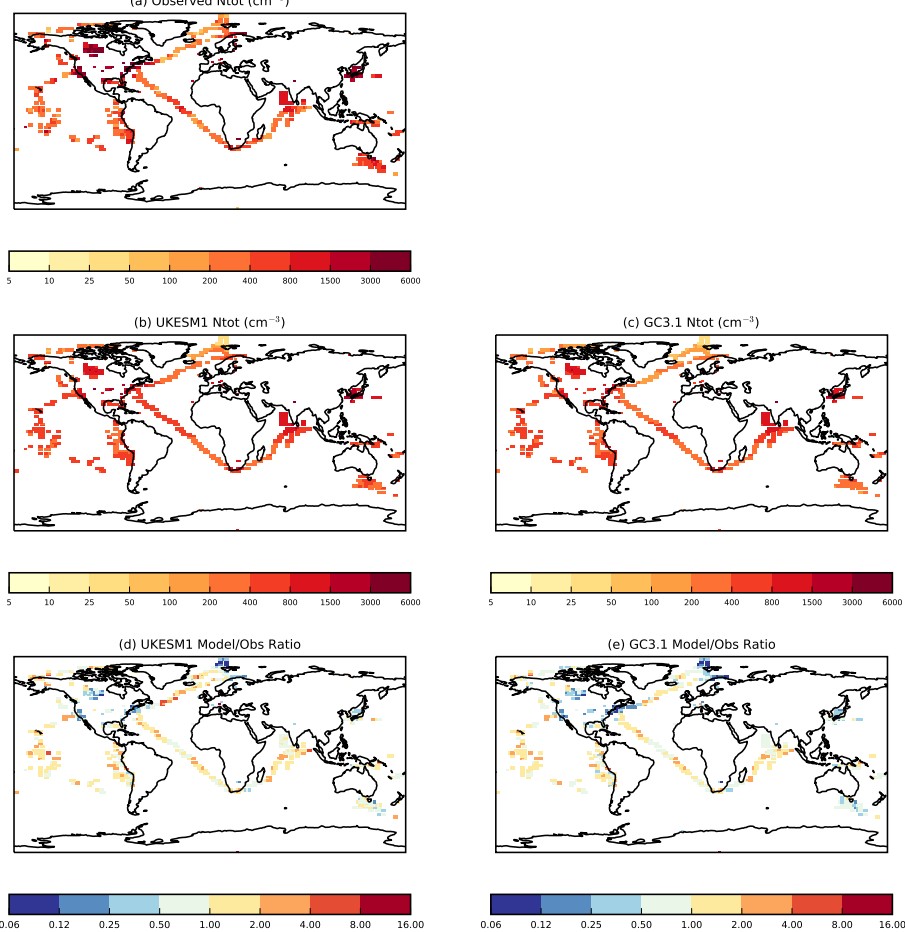

**Figure 11.** Comparison of simulated $N_{tot}$ (total particle concentration with diameter > 3 nm) $(cm^{-3})$ from (a) UKESM1 and (b) GC3.1 against (c) gridded observations from a combinations of ground-based, ship and aircraft campaigns. The ratio of model to observed data for (d) UKESM1 and (e) GC3.1.

In contrast the difference in $N_d$ between UKESM1 and GC3.1 is much smaller, with differences in the region of 3-4% over the ocean and 2% over land. UKESM1 has higher $N_d$ than GC3.1 over most ocean basins but differences are marginal over land. The small global mean change over land ($185.36\,cm^{-3}$ versus $186.74\,cm^{-3}$) is the result of compensating differences in the Northern and Southern Hemispheres (Figure 13c) where UKESM1 has higher $N_d$ over land in the Northern Hemisphere but lower $N_d$ in the Southern Hemisphere. However, these regional differences remain less than 10%.

The averaged $N_d$ over land and ocean in both models is in good agreement with the Grosvenor $N_d$ in particular and the models generally capture the elevated $N_d$ over and downwind of anthropogenic source regions as well as elevated $N_d$ in the main stratocumulus regions off the western coasts of California, Namibia and Chile (Figure 12c and d) . However clear systematic biases in the simulated spatial distributions are apparent (Figure 13a and b) and despite the large difference in

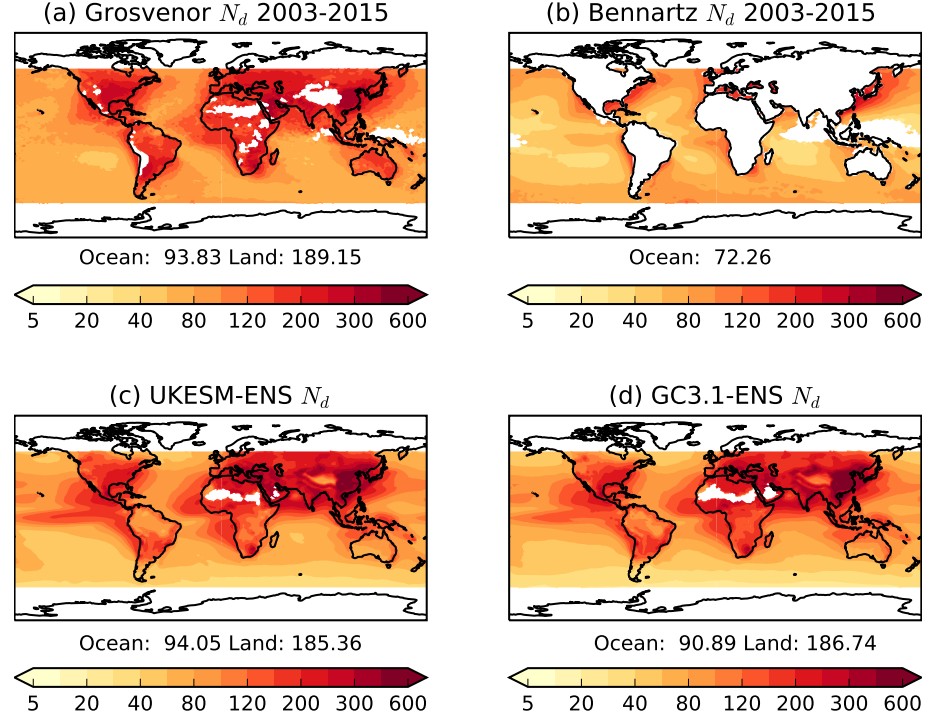

**Figure 12.** Observed and simulated annual mean $N_d$ ($cm^{-3}$) from years 2003–2014 from (a) Grosvenor et al. (2018b) MODIS product, (b) Bennartz and Rausch (2017) MODIS product and (c) UKESM1 and (d) GC3.1 simulations.

the satellite $N_d$ the model biases are generally consistent against both observational products. The models underestimate high latitude $N_d$ and overestimate $N_d$ in the marine stratocumulus and also in the trade and shallow cumulus regimes. Recent updates to the aerosol activation scheme documented in (Mulcahy et al., 2018) improved the representation of aerosol activation in convective cloud regimes and this generally reduced $N_d$ in the tropics. Biases in convective cloud $N_d$ should be interpreted

with caution given the large observational uncertainty associated with these cloud types (see Section 4.4) with the Grosvenor $N_d$ for instance not retrieving $N_d$ in clouds with tops higher than $3.2\,km$.

  The higher $N_d$ in UKESM1 at high latitudes improves the model bias relative to GC3.1. A further assessment of simulated $N_d$ over the Southern Ocean and the impact of including the PMOA source is detailed in the next Section. Lower $N_d$ over the Indian Ocean in UKESM1, potentially due to the lower DMS emissions in these regions, act to reduce the positive bias relative

to GC3.1 but UKESM1 has a larger positive bias in the marine stratucumulus regions.

  Over land, UKESM1 underestimates over most continents apart from Asia and parts of North Africa where $N_d$ is overestimated (Figure 13a). The lower land $N_d$ in GC3.1 in the Southern Hemisphere will improve the bias in GC3.1 in this region but degrade the bias over North America and China. Differences in $N_d$ between the models over land will be due to a combination of factors including differences in natural emissions of terrestrial biogenic sources, aerosol scavenging and aerosol

activation processes. The aerosol activation over land surfaces will be influenced by the boundary layer turbulent kinetic energy



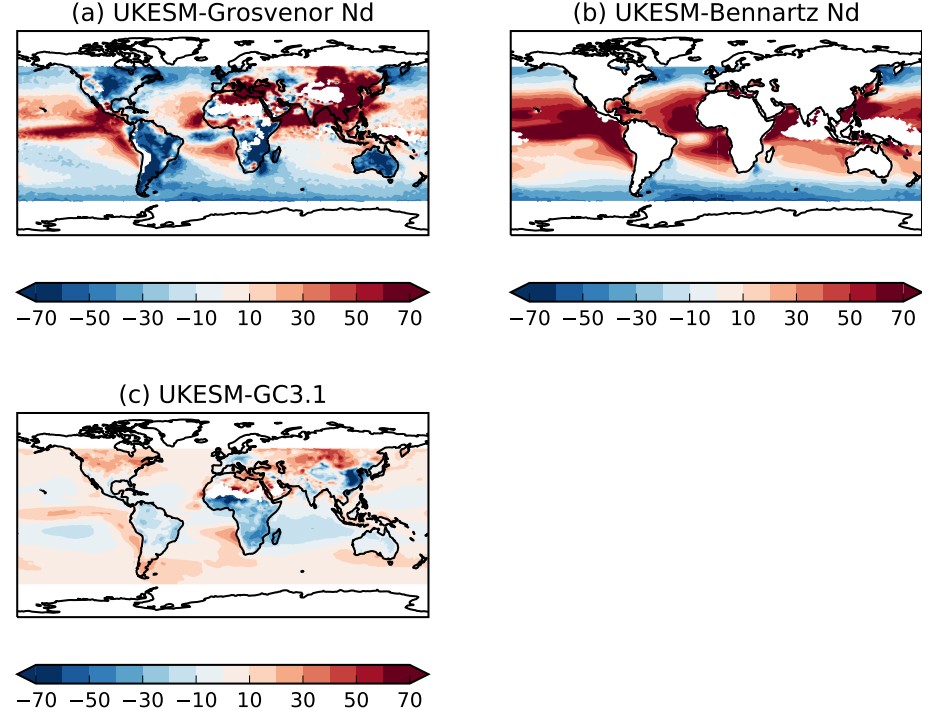

**Figure 13.** Annual mean bias in UKESM1 $N_d$ from years 2003–2015 compared with (a) Grosvenor et al. (2018b) MODIS product, (b) Bennartz and Rausch (2017) MODIS product. (c) UKESM1-GC3.1 $N_d$.

flux which determines the sub-grid variability of the updraught velocities. Given the differences in the land surface properties of UKESM1 and GC3.1 we expect differences in the turbulent mixing which will also impact the vertical distribution of the aerosol.

### 5.6 Evaluation of primary marine organic aerosol

5 The emission of PMOA is a new source of marine aerosol in UKESM1. We therefore explore and evaluate the impact of this additional source of OM in the model and in comparison with GC3.1 which instead scales the marine DMS emission as a proxy for this missing source (Mulcahy et al., 2018). We focus our evaluation on the Southern Ocean region due to the high occurence of pristine air masses in this region (Hamilton et al., 2014) and therefore a low risk of the OM mass concentrations being contaminated by anthropogenic emissions.

10 The annual average global emission of PMOA is $4.5\,\mathrm{Tg\,yr^{-1}}$ during the present-day period evaluated here (Figure 14). This is in good agreement but slightly below the global PMOA emission rate of $6.3\,\mathrm{Tg\,yr^{-1}}$ found in Gantt et al. (2012) who use the same emission parameterization but apply an additional global scaling factor of 6 in Equation 3. The value is also within the range of emission rates calculated using different emission parameterizations in the same model (Gantt et al., 2012) and in numerous other modelling studies which span a range of $2$–$70\,\mathrm{Tg\,yr^{-1}}$ (see Gantt and Meskhidze (2013) for a review)

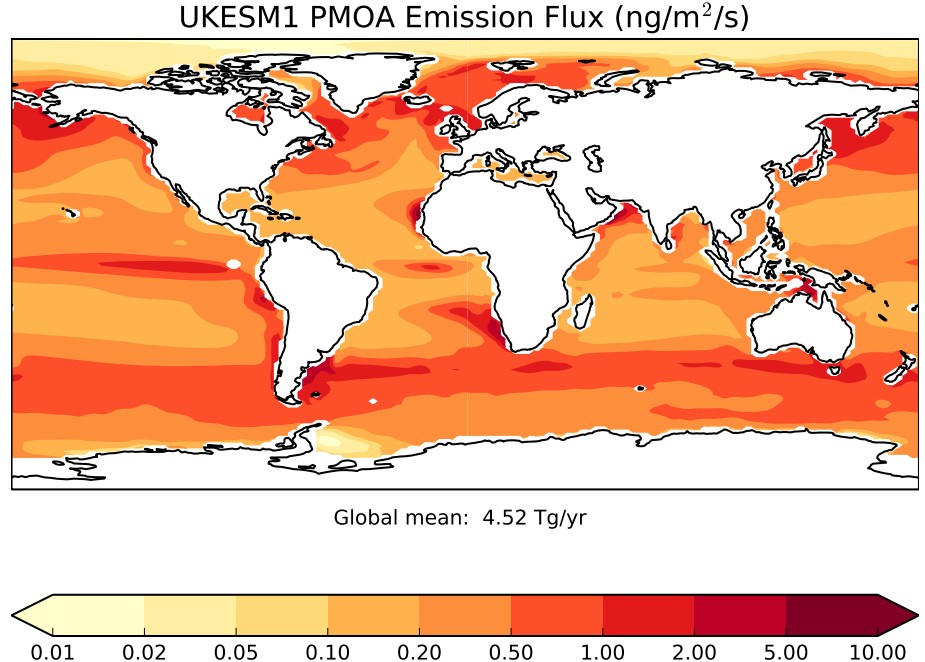

Global mean:  4.52 Tg/yr

**Figure 14.** Annual mean emission of primary marine organic aerosol from UKESM1-AMIP in $\mathrm{Tg[OM]/yr}$.

but with many studies showing values in the region of $10\,\mathrm{Tg\,yr^{-1}}$ (Spracklen et al., 2008; Lapina et al., 2011; Vignati et al., 2010). Scaling the PMOA emissions by 6 in Gantt et al. (2012) was found to lead to improved agreement of surface mass concentrations at Mace Head ($53.33\,^\circ\,\mathrm{N},9.90\,^\circ\,\mathrm{W}$) and Amsterdam Island ($37.80\,^\circ\,\mathrm{S},77.57\,^\circ\,\mathrm{E}$) . Apart from the scaling factor other model-dependent differences will impact the $OM_{SSA}$ such as differences in sea-salt emissions, $10\,\mathrm{m}$ windspeeds and the

use of different Chl-a data. Comparing the spatial distribution of the PMOA emissions with test simulations where Equation 2 was driven with observationally-based Chl-a from GlobColour (Ford et al., 2012) shows a strong dependence of the emission parmeterization on the underlying chl-a (not shown).

    In order to evaluate the PMOA we run an additional UKESM1-AMIP simulation where we deactivate the PMOA emission source. This "NoPMOA" simulation is required as we don't track the PMOA as a separate tracer in the model but add the

emitted mass and number to the existing OM tracers which will, in addition, be composed of both anthropogenic and terrestrial OM components. Thus two separate runs are required to allow us to determine the contribution of PMOA to the simulated OM mass, number and subsequently $\mathrm{N_d}$.

    Figure 15a compares the seasonal cycle of simulated OM surface mass concentrations at Amsterdam Island with observations from Sciare et al. (2009). The observations show a clear seasonal cycle in organic aerosol concentrations peaking in the austral

summer. GC3.1 and NoPMOA underestimate the observations in all months and have a much weaker, inverse seasonal cycle with peak emissions occuring in winter. Such low concentrations are consistent with a lack of biogenic or local anthropogenic sources in this pristine remote location. When PMOA is included in the model the low bias in the OM concentrations is clearly



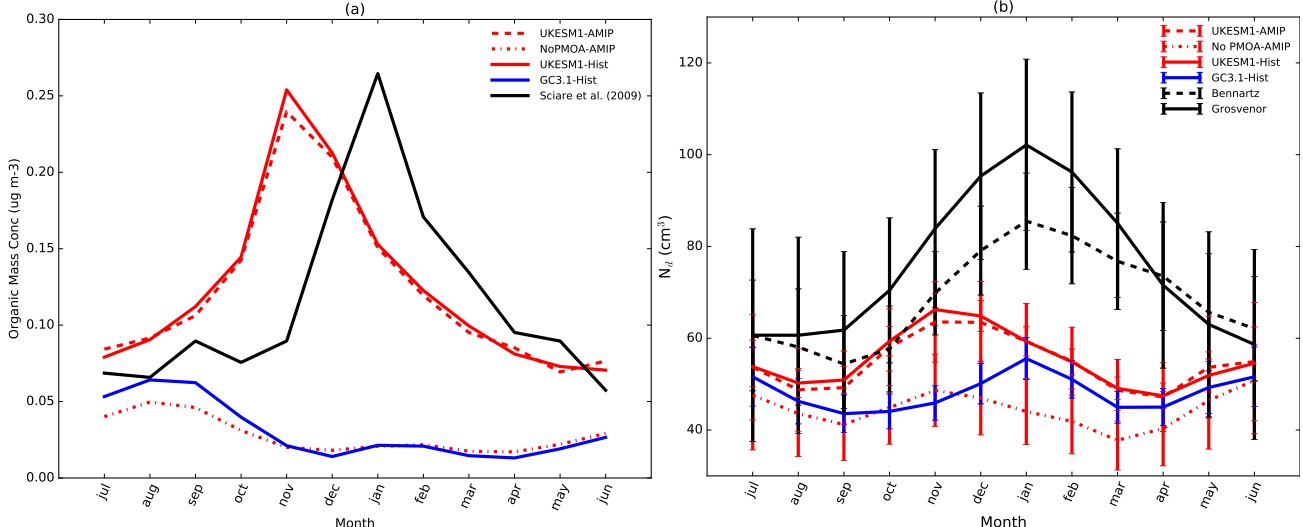

**Figure 15.** Seasonal evaluation of simulated (a) OM surface mass concentration at Amsterdam Island and (b) $N_d$ averaged averaged over $30°$S-$70°$S from UKESM1-AMIP, a UKESM1-AMIP simulation with no PMOA source and the UKESM1 and GC3.1 historical ensemble means. The OM mass observations in (a) are taken from Sciare et al. (2009). The $N_d$ observations used in (b) are taken from the Grosvenor et al. (2018b) and Bennartz and Rausch (2017) satellite $N_d$ products. The simulated monthly $N_d$ data is taken from the years 2003 to 2014 inclusive to match the satellite temporal coverage. Given the low data coverage of satellite retrievals in the austral winter the UKESM1 and Bennartz data has been spatially sampled according to the Grosvenor data product

improved in all months. The model also now exhibits a much stronger seasonal cycle which captures the magnitude of the summer peak, although the peak emission occurs slightly earlier in the model compared to the observations (November versus January). The excellent agreement between the UKESM1 historical ensemble mean and the UKESM1-AMIP offers confidence that use of the atmosphere-only configuration to assess the impact of PMOA is appropriate in this case.

UKESM1 with PMOA shows a clear improvement in the seasonal cycle of $N_d$ over the Southern Ocean compared with MODIS retrieved $N_d$ with peak monthly mean $N_d$ increasing by up to $20\,\mathrm{cm}^{-3}$ relative to the NoPMOA simulation (Figure 15b). When PMOA is included the simulated $N_d$ is within the observed variability from the austral winter through to the early summer but underestimates the peak summer $N_d$. Similar to the seasonal cycle in OM concentrations, the seasonal peak in $N_d$ occurs about a month too early. GC3.1 underestimates the observed $N_d$ in all months and has $N_d$ values which are consistently

lower than UKESM1. The largest underestimation of the observed $N_d$ (of up to $50\,\mathrm{cm}^{-3}$) occurs during the summer months although the peak $N_d$ does occur in the correct month (January). Interestingly, the NoPMOA simulation has a much smaller seasonal cycle in $N_d$ than GC3.1 likely due to the much lower DMS emissions in UKESM1.

     Figure 16 shows the seasonal cycle in Chl-a and DMS surface concentrations simulated by UKESM1 and from corresponding observation-based climatologies over the Southern Ocean region (Ford et al., 2012; Lana et al., 2011). The Lana et al. (2011)

DMS seawater climatology being what is used in GC3.1. It should be noted that the UKESM1 Chl-a in Figure 16 is $0.5 \times$ Chl-a



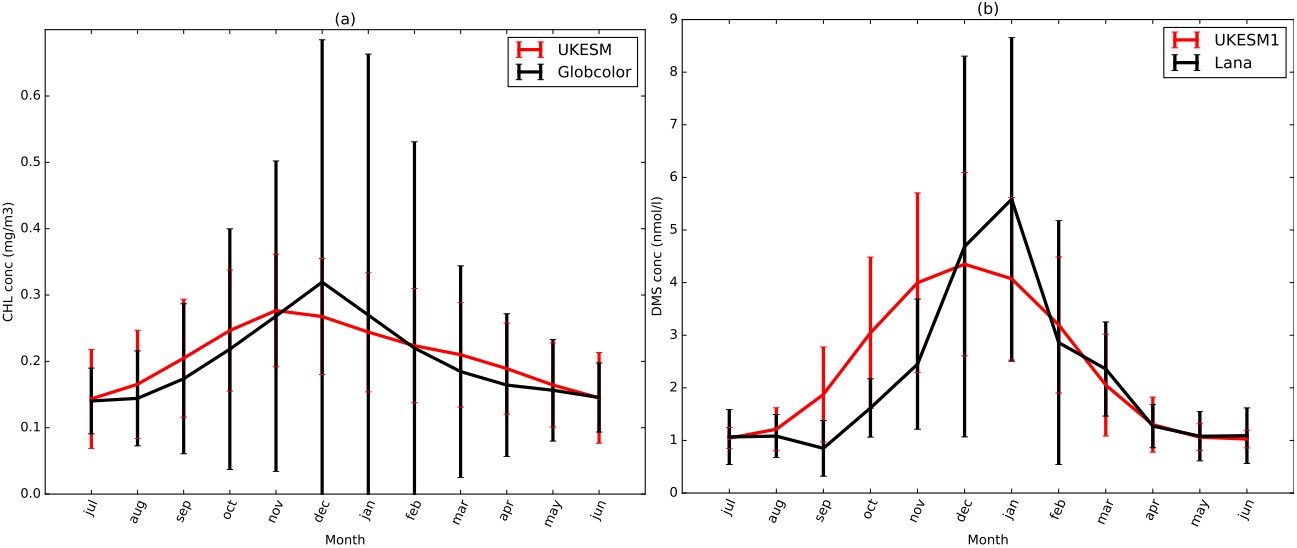

**Figure 16.** Seasonal comparison averaged over 30S-70S of (a) ocean surface Chl-a concentration and (b) seawater DMS concentration simulated interactively from UKESM1 across $30\,^{\circ}$S-70$\,^{\circ}$S. Simulations here are from UKESM1-AMIP and a parallel run with PMOA emissions excluded. Given the low data coverage of satellite retrievals in the austral winter the UKESM1 and Bennartz $N_d$ has been spatially sampled according to the Grosvenor $N_d$ product. The surface Chl-a plotted in (a) and used in Eqn 2 is $0.5 \times$ Chl-a simulated by the MEDUSA ocean biogeochemistry model, see main text for details.

simulated by the ocean biogeochemistry model, MEDUSA. As already stated this is due to the general overprediction of this variable in MEDUSA, particularly in the Southern Ocean (Yool et al., 2013). Overall, the scaled monthly Chl-a in UKESM1 agrees well with Globcolor Chl-a in this region when scaled and while small biases exist the mean monthly Chl-a is within 1 standard deviation of the observations. It is worth noting that the simulated seasonal peaks in both OM surface concentration

5    and $N_d$ in Figure 15 are correlated with the peaks in simulated Chl-a but not DMS concentrations demonstrating that the seasonal cycle of $N_d$ in this region is controlled largely by the organic aerosol in the model. The observations suggest however that the seasonal peak in $N_d$ is driven by DMS than Chl-a although a lagged response to Chl-a is possible (Rinaldi et al., 2013).

While the UKESM1 Chl-a peaks in November and DMS peaks in December, the observations show a peak in December and January for Chl-a and DMS respectively. UKESM1 simulated DMS is overpredicted in spring, underpredicted in summer and

10   agrees well with observations in the autumn (16). The inability of the simulated Chl-a and DMS to capture the correct seasonal cycle highlights some deficiencies in the ability of the ocean biogeochemistry model, MEDUSA, to capture the complex biological productivity in the Southern Ocean (Yool et al., 2019, 2013).





# 6 Discussion

UKESM1 and GC3.1 offer a unique opportunity to explore and improve understanding of aerosol-climate interactions through the exploitation of a traceable hierarchy of global climate models. These two models employ the same physical atmosphere-ocean components and in essence the same aerosol scheme but differ in their level of interaction with the full Earth system and
in the specification of a small number of other aerosol properties, most notably the inclusion of an additional organic aerosol source over the ocean in UKESM1. A number of notable differences in the simulation of aerosols between the models have been highlighted in the current study. This paper attempts to characterize the overall climatology of aerosol and aerosol-cloud properties in both models with the aim of facilitating a broad understanding of the key drivers of the underlying differences and associated model uncertainties. Subsequent future analysis will consider in more detail interactions between simulated marine
and terrestrial biogeochemistry, atmospheric chemistry and aerosol properties in UKESM1.

The additional ES components in UKESM1 add complexity in particular with respect to aerosols. Coupling of the aerosol emissions and chemistry to dynamic vegetation, ocean biogeochemistry and a complex chemistry scheme introduces extra degrees of freedom in fully coupled ES models. This leads to the potential for biases in the interactively simulated processes where in GC3.1 they are prescribed, in most cases from present-day observational based climatologies. The GC3.1 treatment
of emissions and chemistry does not allow for future changes in these variables, for instance climate feedbacks on ocean productivity influencing marine emissions of DMS or ozone depletion influencing the aerosol oxidation pathways. Therefore one might expect smaller model biases in GC3.1 given the present-day nature of the evaluation presented here, although in many instances we find this is not the case.

The inclusion of an interactive emission scheme for BVOCs enables the BVOC emissions to change in response to changes
in land use and climate over the industrial period. This is found to lead to improvements in the simulation of organic aerosol over North America. The nature of the aerosol chemical oxidation is also found to be important with oxidation of monoterpene by $NO_3$ for instance, significantly higher in GC3.1 due to lack of a removal mechanism for this species. The different aerosol chemistry also leads to notable differences in the aerosol sulphur cycle. Despite nearly double the emissions of marine DMS in GC3.1 the annual mean sulphate loads are comparable. This is due to the different oxidation and scavenging lifetimes in
the models. The former is likely driven by a combination of different oxidants as well as differences in the DMS chemistry. Limitations in the offline oxidant scheme are apparent where perpetual sources of oxidants could significantly influence the amount of aerosol produced in oxidant limited regions. This has potentially important implications for the aerosol forcing as recently highlighted by Karset et al. (2018).

While dust emissions in UKESM1 are more than double the emissions in GC3.1, the relative increase in dust burden is
much smaller (less than 35%). The different tuned parameters for dust employed in UKESM1 is a balance between achieving an optimal performance of dust metrics such as surface dust concentrations and aerosol optical depth against present day observations and achieving a realistic dust size distribution which has impacts for the remote transport of dust and subsequent deposition into the ocean in the fully coupled UKESM1. Recent observational studies (Ryder et al., 2019, 2013) support the





existence of giant dust particles close to source and highlight the potential important longwave and shortwave radiative effect associated with such large dust particles.

Biogenic emissions from the ocean are believed to be responsible for the observed seasonal cycle in aerosol and consequently $N_d$ in the Southern Ocean (Behrenfeld et al., 2019; Sanchez et al., 2018; McCoy et al., 2015; Meskhidze and Nenes, 2006)

and also in clean marine regions such as the North Atlantic during periods of high ocean productivity. Biogenic sources include DMS, methanesulphonoic acid (MSA) as well as marine organics. Uncertainty in the CCN from these sources has large implications for aerosol forcing in remote marine regions (Carslaw et al., 2017; Moore et al., 2013). The implementation of PMOA in UKESM1 clearly brings improvements in terms of the seasonal cycle of organic aerosol mass and $N_d$ in the Southern Ocean. However an underestimation of $N_d$ still remains. Given our global PMOA emissions are to the lower end of

published ranges discussed above one could argue for the inclusion of a global scaling factor as used in Gantt et al. (2012). We adopt a conservative approach here in order to balance the low bias in $N_d$ with the impact on top-of-atmosphere radiation biases. Furthermore, the good agreement of OM mass with observations at Amsterdam Island suggests an alternative source of error, possibly a low bias in the underlying DMS concentration in summer. The apparent low sensitivity of $N_d$ to DMS shown in Figure 15 is inconsistent with a previous study (Korhonen et al., 2008) which, using a sectional version of GLOMAP,

highlight a large seasonal cycle in CCN in the Southern Ocean controlled largely by DMS. Global DMS emissions in UKESM1 are certainly on the lower end of the likely range of emissions (Lana et al., 2011) and are lower than GC3.1 due to the lower DMS seawater concentrations simulated by the MEDUSA ocean biogeochemistry model employed in UKESM1. Furthermore, Korhonen et al. (2008) use the Nightingale et al. (2000) parameterization for the air-sea emission flux of DMS while both UKESM1 and GC3.1 use the parameterization of Liss and Merlivat (1986). The emitted flux in Nightingale et al. (2000) varies

as a function of $U_{10}^2$ in comparison to the linear dependence on $U_{10}$ in the Liss and Merlivat (1986) parameterization. The stronger windspeed dependence will lead to higher DMS emissions particularly at high windspeeds when the Nightingale et al. (2000) parameterization is used. The use of Liss and Merlivat (1986) is supported by recent direct measurements of DMS air-sea exchange (Yang et al., 2011) and studies which show that the high solubility of DMS results in lower air-sea transfer velocities at high winds compared to less soluble gases like $CO_2$ (Bell et al., 2013; Wanninkhof, 2014). Revell et al. (2019)

highlight further possible biases in the simulation of sea salt and role of DMS chemistry in aerosol biases in the Southern Ocean. Our evaluation of AOD and $N_d$ is in agreement with Revell et al. (2019) who use a variant of the UKESM1 and GC3.1 atmosphere model, GA7.1, with the full StratTrop chemistry scheme so further improvements in these areas will be investigated in the future.

Over the Northern Hemisphere continents, in regions where the simulated surface mass concentrations is generally under-

estimated the total aerosol number concentration and number concentration of particles greater than 50 nm are also underestimated. Underestimation of AOD is not surprising given the aerosol model is missing aerosol species such as nitrate, however underestimation of the surface $SO_4$ aerosol mass is also found. The evaluation of the size segregated number concentrations highlights a potential overestimation of both $N_{50}$ and $N_{tot}$ over most ocean regions, with the exception of high latitude oceans. This issue appears to be worse in UKESM1 due to the higher number concentrations in all size modes in UKESM1. This is

believed to be driven by difference in the treatment of natural aerosol, different oxidation and scavenging lifetimes, but also



aerosol higher nucleation rates in UKESM1 which increase nucleation mode number concentrations which can subsequently grow via condensation and coalescence to larger CCN sizes. Model differences in $N_d$, the variable most important for the aerosol-cloud forcing, are relatively small and are typically less than 10%. Indeed over ocean, where the satellite data is most reliable, both models underestimate $N_d$ in high latitude regions but have a positive bias in the marine stratocumulus cloud

regimes which are consistent with an overestimation of $N_{50}$ found in Californian stratocumulus clouds.

There are a number of caveats pertaining to the evaluation of the aerosol number concentration and $N_d$, most importantly the lack of representative measurements on the global scale. The evaluation of aerosol number presented in this study comprises observations from a large number of measurement campaigns compiled by GASSP as well as ground-based measurements. Use of campaign data which are often targeting specific physical processes or aerosol regimes may not be appropriate for

evaluation of a global climate model but to-date this remains the most comprehensive dataset available (Reddington et al., 2017). Similarly, satellite retrievals of $N_d$ have large uncertainties and have not been fully validated in different cloud regimes. Having globally representative aerosol measurements is essential in order to constrain global aerosol microphysical processes and subsequent aerosol forcing in models. Long-term monitoring networks are predominantly close to source with often sparse aerosol information in remote oceans regions which are often key regions of aerosol forcing uncertainty (Regayre et al., 2018).

Notwithstanding these limitations, the overall simulation of aerosol in the historical free-running climate simulations of both models compares remarkably well with the observations in this study.

## 7  Conclusions

The aerosol scheme employed by the physical and Earth system models, HadGEM3-GC3.1 and UKESM1, is documented in detail. Differences in the aerosol representation relate to the interactive simulation of the natural aerosol emissions in UKESM1,

including dust, DMS and terrestrial BVOCs as well as the inclusion of a new marine organic aerosol source replacing the scaled marine DMS emissions in GC3.1. The impact of these differences on the aerosol distributions are fully characterized and a detailed evaluation of the present-day period of the historical CMIP6 simulations is conducted.

Overall, both models compare well with observations and capture the global spatial distributions in AOD and cloud droplet number concentrations. Some regional biases are noted including an overestimation of $N_d$ in the marine stratocumulus cloud

regimes and an underestimations of aerosol optical depth in dust-dominated regions. Regional trends in surface sulphate concentrations are well represented in the models although they generally tend to underestimate the absolute magnitude of the sulpate concentrations over Europe and the eastern US while overestimations are apparent over the western US are found. The inclusion of the interactive BVOC emission scheme and marine organic aerosol source in UKESM is found to improve surface mass concentrations of organic aerosols. The inclusion of marine organic aerosol is furthermore found to improve the seasonal

cycle of cloud droplet number concentration in the Southern Ocean although biases associated with the interactive simulation of DMS and Chl-a in UKESM1 are evident.

UKESM1 is believed to be one of the most comprehensively coupled models (in terms of the number of Earth system interactions) contributing to CMIP6, in particular with respect to aerosols. It is therefore highly encouraging that such interactions



with the terrestrial and ocean biogeochemical and atmospheric chemistry systems not only do not degrade present-day model performance but in many instances improve the present-day comparison against observations. This builds confidence in the use of this model in the wide-ranging forcing and feedback experiments being conducted as part of CMIP6 and the potentially important role of aerosol in modulating or amplifying future climate feedbacks. While GC3.1 also compares well on the whole

against observations, limitations with respective to the simplified chemistry scheme employed and the representation of natural aerosol sources are evident. The implications of such ES interactions on the aerosol forcing will be explored in more detail in a future study.

*Code and data availability.* Due to intellectual property rights restrictions, we cannot provide either the source code or documentation papers for the UM or JULES.

*Obtaining the UM.* The Met Office Unified Model is available for use under licence. A number of research organisations and national meteorological services use the UM in collaboration with the Met Office to undertake basic atmospheric process research, produce forecasts, develop the UM code, and build and evaluate Earth system models. For further information on how to apply for a licence, see http://www.metoffice.gov.uk/research/ modelling-systems/unified-model (last access: 13 December 2019).

*Obtaining JULES.* JULES is available under licence, free of charge. For further information on how to gain permission to use JULES for

research purposes see http://jules-lsm.github.io/access_req/JULES_access.html (last access: 13 December 2019).

*Data availability.* The simulation data used in this study are archived on the Earth Sytem Grid Federation (ESGF) node (https://esgf-node.llnl.gov/projects/cmip6/). The model Source IDs are HadGEM3-GC31-LL for HadGEM3-GC3.1 and UKESM1-0-LL for UKESM1. The HadGEM3-GC3.1 historical simulations are identified by the following Variant Labels: r1i1p1f3 to r4i1p1f3. UKESM1 historical simulations are identified by the following Variant Labels: r1i1p1f2 to r4i1p1f2, r5i1p1f3 to r7i1p1f3, and r8i1p1f2 to r9i1p1f2. We ac-

knowledge the use of MODIS AOD data from https://earthdata.nasa.gov and ESA CCI AOD data from http://www.esa-aerosol-cci.org. AOD data from AERONET is available from https://aeronet.gsfc.nasa.gov/. Aerosol number concentration data is available through the GASSP project (http://gassp.org.uk) and the EBAS database (http://ebas.nilu.no). Aerosol mass concentrations are available from the EMEP (http://ebas.nilu.no/), IMPROVE (https://views.cira.colostate.edu/fed/), and EANET (https://www.eanet.asia/product/index.html) databases. Information on the EMEP network can be found in Tørseth et al. (2012).

*Author contributions.* JPM led the analysis and wrote the manuscript. CJ, CGJ, AP, CES, AS, STT, MTW provided significant text, analysis and/or figures presented in this manuscript. JPM, CJ, CGJ, AS, MTW, NLA, NB, JB, KSC, MD, DG, MG, CH, BJ, AJ, ZK, GM, JM, FO'C, CR, MR, NAJS, PS, MS and SW made significant contributions to the development and/or evaluation of the aerosol schemes in the model configurations evaluated in this study. JPM, CGJ, AS, GAF, RH, MS, JP, YT and AY made significant contributions to the development and implementation of the Earth system aerosol couplings in UKESM1. MA ran the HadGEM3-GC3.1 historical and AMIP simulations. STR

ran the UKESM1 historical and AMIP simulations. JW led the data processing of the simulation outputs. Additional simulations were run by JPM. All co-authors contributed invaluable comments on this manuscript.





*Competing interests.* The authors declare that they have no conflict of interest.

*Acknowledgements.* JPM, CJ, AS, STT, MA, MD, GAF, MG, CH RH, BJ, AJ, FOC, YT, JW and SW were supported by the Met Office
Hadley Centre Climate Programme funded by BEIS and Defra (GA01101). CJ, CGJ, AP, JP, STR and AY were supported by the National
Environmental Research Council (NERC) National Capability Science Multi-Centre (NCSMC) funding for the UK Earth System Modelling

project, grant number NE/N017951/1. CGJ, CES, GAF, FOC, JP, and AY additionally acknowledge the EU Horizon 2020 CRESCENDO
project, grant number 641816. MTW is supported by the Earth System and Climate Change Hub of the Australian Government's National
Environmental Science Programme (NESP). NLA acknowledges support from NERC and NCAS for funding aspects of UKCA model
development and evaluation. ZK acknowledges the support of a Met Office CASE studentship; the following NERC projects, AEROS
[NE/G006148/1]; GASSP [NE/J022624/1] and the European Research Council (ERC) project ACCLAIM [FP7-280025]. PS acknowledges

funding from NERC projects NE/J022624/1 (GASSP), NE/L01355X/1 (CLARIFY), NE/M017206/1 (IMPALA) and NE/P013406/1 (A-
CURE) as well as from the ERC project RECAP under the European Union's Horizon 2020 research and innovation programme with grant
agreement 724602. STT would like to acknowledge the UK-China Research and Innovation Partnership Fund through the Met Office Climate
Science for Service Partnership (CSSP) China as part of the Newton Fund. Much of the model development work made use of MONSooN
and Monsoon2, a collaborative High Performance Computing facility funded by the Met Office and the NERC. This work used JASMIN, the

UK collaborative data analysis facility.

For making their measurement data available to be used in this study we would like to acknowledge the EMEP (http://ebas.nilu.no/), IM-
PROVE (https://views.cira.colostate.edu/fed/), and EANET (https://www.eanet.asia/product/index.html) measurement networks along with
any data managers involved in data collection. IMPROVE is a collaborative association of state, tribal, and federal agencies and international
partners. The U.S. Environmental Protection Agency is the primary funding source, with contracting and research support from the National

Park Service.

Ground station observations of aerosol number concentrations were collated via GASSP (Reddington et al., 2017) and public data available
on the EBAS database (http://ebas.nilu.no). The authors thank the many scientists who have contributed aerosol data from both observational
sites and campaigns that went into this evaluation and are described in detail in Appendix B of Johnson et al. (2019b). The EBAS database
has largely been funded by the UN-ECE CLRTAP (EMEP), AMAP and through NILU internal resources. Specific developments have been

possible due to projects like EUSAAR (EU-FP5; EBAS web interface), EBAS-Online (Norwegian Research Council INFRA; upgrading of
database platform) and HTAP (European Commission DG-ENV; import and export routines to build a secondary repository in support of
www.htap.org). A large number of specific projects have supported development of data and meta data reporting schemes in dialog with data
providers (EU; CREATE, ACTRIS and others). For a complete list of programmes and projects for which EBAS serves as a database, please
consult the information box in the Framework filter of the web interface. These are all highly acknowledged for their support.





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
