# Peer review of "Description and evaluation of aerosol in UKESM1 and HadGEM3-GC3.1 CMIP6 historical simulations"

_Geoscientific Model Development, 2019_

## Referee Comment (RC1) · Anonymous Referee #1 · 8 Apr 2020

General comments:

This paper document the GLOMAP aerosol scheme as implemented in the Earth System Model UKESM1 and the corresponding, but less advanced, physical model HadGEM3-GC3.1. It also evaluates the aerosol and cloud droplet properties in the present-day period of the historical simulation of CMIP6 and fixed-SST simulations to several observational datasets. With minor modifications, the paper will be a very useful reference for later studies applying these models to aerosol related questions and for studies analyzing their CMIP6-performances.

Specific comments:

[Figure]

- Natural emissions: even though the citations leads to detailed descriptions of the emission algorithms, it would be beneficial for the reader to be served mathematical expressions for more of the emissions to easier see how they vary with climate variables such as temperature and wind.

- BVOC: the models only include SOA production from monoterpenes. Although these emissions are scaled up, they are different in nature compared to other BVOCs. A few sentences about the disadvantages of this simplification would be nice.

- Oxidation: several of the differences in behavior between the two models are linked to their different treatment of the oxidants and the following differences in oxidation close to and away from emission sources. These statements will benefit from map plots showing horizontal distribution of the oxidants and/or horizontal distribution of loss of precursors through the chemical processes.

- Ice crystals: Cloud droplet number concentration is thoroughly evaluated, but what about the ice crystals? If the model does not include aerosol impact on cold clouds, it should be mentioned. If some of the aerosols can act as INP, that should also be described.

- Dust aerosol scheme: please add a sentence explaining why dust is not implemented in the regular aerosol scheme, but needs a single aerosol scheme.

- Cloud droplet number concentration: Can you add some sentences arguing why the satellite datasets are comparable to your model output? Do you use the same criteria for your modeled Nd as in the satellite products (for example cloud fractions > 80 %). With satellites only seeing cloud tops and specific clouds under specific conditions, the model output would be more comparable if the same criteria where used, or if a satellite simulator was applied. If a satellite simulator was applied, it should be mentioned. If not, it should be mentioned that the model output would be more comparable to the satellite products if a satellite simulator was used.

[Figure]

- The conclusion would benefit from adding a few sentences about which improvement that should be made to future model versions.

Technical corrections:

p2, l19: "on which cloud droplets can form" → "on which cloud droplets and ice crystals can form"

p2, l20: "increase anthropogenic emissions leads to increases" → "increase anthropogenic emissions usually leads to increases" (the opposite can occur if it's already many aerosols and low supersaturation)

p9, Eq. (2): missing two end parentheses.

p11, Table 2: COS is not explained in the text. Please do so.

p11, Table 2: What happens with MSA?

p12, Table 3: be consequent regarding SEC_ORG and Sec_Org

p13, l10: add space after .

p13, l18, add space after .

p13, l24: add space before nm

p20, l15: "historican" → "historical"

p21: Table 4 and Table 5 can be merged for easier comparison of the models by using one type of brackets for UKESM, and another type of brackets for HadGEM

p22, Figure 1. "mg[SO2]m$^2$" → "mg[SO2]/m$^2$"

p22, l23: remove . Before "for all years"

p23, Figure 2: Add one column with differences between the models.

p24, Figure 3: Is the regression correct? It does not look correct for some of the figures.

[Figure]

The figure will also be better with partly transparent dots.

p26, Figure 5: "netorks" → "networks"

p33-34, Figure 10 and 11: add arrows on the colorbar (like the other figures in the paper)

---

## Referee Comment (RC2) · Anonymous Referee #2 · 7 Jun 2020

Aerosol modeling is a major challenge, in particular aerosol-cloud interactions. This work evaluates the performance of the GLOMAP-mode modal aerosol scheme when implemented in the GCM HadGEM3-GC3.1, and the larger ESM UKESM1 which is built on top of that GCM. The authors provide a comprehensive description of relevant differences between the two models. Using an ensemble of results from each model, they describe relevant differences in the aerosol simulations by each model when using GLOMAP, performing extensive comparison to observations. They also evaluate the effect of implementing a parameterization for marine primary organic aerosol emissions into UKESM1.

[Figure]

The central question of this work is interesting – in particular the question of how much additional skill in aerosol representation is achieved by using a more comprehensive earth system model. The approach taken to address this question is to compare the results of ensemble simulations with GC3.1 to those with UKESM1. The authors' conclusions – that the more complex UKESM1 does indeed produce a more accurate simulation of aerosol burdens, or at least does not produce a worse simulation - are mostly supported by their data, with some caveats.

The comparisons that the authors perform are pertinent and thorough, looking in detail at the quantities relevant to the evaluations in question. Their investigation of the effects of a new marine primary organic aerosol source, for example, is appropriate, and the observational sets used to evaluate the models are fitting. However, the paper suffers from two significant drawbacks.

Firstly, the analysis appears to mostly rely on comparison of completed ensembles, with few examples of dedicated simulations which isolate a specific process present in one model but not the other. This makes it difficult to evaluate the true contribution of individual differences between the UKESM1 and GC3.1 to the observed differences. This undermines the authors' claims regarding the value of having a "traceable hierarchy". For example, it is difficult to fully interpret differences in sulphate loading (or to evaluate the accuracy of the authors' interpretations) because the two models use different approaches to distribute sulphur emissions. To truly take advantage of the "traceable hierarchy", more sensitivity simulations – such as the one performed here for evaluation of the effect of the primary marine organic aerosol emissions parameterization – would be needed, and would have significantly improved the paper. Without them, it is difficult to say why exactly one model is more accurate than the other. This mostly constitutes a missed opportunity and a reduction in the paper's utility.

Secondly, the comparisons made between the two models are mostly qualitative and subjective. In spite of an abundance of numerical output, the majority of comparisons made by the authors – both between the two models and against observations – are

qualitative. This seems unnecessary, and I strongly recommend that the authors revisit their results and discussion sections to make them more quantitative. Differences are frequently said to be "clear", "small", "large", "high", "low", and so on, with comparisons said to be "good" or "excellent", when for all such cases a quantitative assessment should be straightforward. Statements about agreement are also often (albeit not always) vague and qualitative, such as: "BC and OM emissions are in good agreement with other models" (P19, L34); "Mineral dust emissions initially appear high in UKESM1 compared to other models" (P20, L10); "the low bias in the OM concentrations is clearly improved" (P37, L17); "the global dust burden... compares well with other models" (P20, L17); and so on. This last example is strange since it was immediately preceded by a statement that comparison is methodologically difficult. The last paragraph of page 20 is similarly problematic, claiming that "BC burdens compare extremely well" but then making no quantitative assessment. Similar statements such as that the model "compares remarkably well with the observations" (P42, L15-16) do not add to the discussion and seem more like assertions than scientific evaluations. Throughout the results and discussions sections, the numerous qualitative comparisons should be replaced with quantitative comparisons. This will enable the readers to evaluate the quality of the comparison for themselves, rather than relying on the authors' opinion.

These issues are by no means fatal to the paper, and it is worth noting that there are still many quantitative comparisons made. If the authors can generally replace the more qualitative or subjective comparisons with quantitative ones, it will significantly improve the paper while also maximizing its scientific value. Additional sensitivity simulations to isolate the specific contributions of each individual difference between the two models would also help to elevate the paper.

Other major comments

Further to the broader need for quantitative comparisons, several comparisons are made to the AeroCom medians (e.g. P20 L21), but it would be more informative to also

include the AeroCom range if possible.

How much of the difference in observed outcomes for sulphate can be explained by the different sulfur dioxide emission approaches? In particular, the lifetime of sulphate aerosol is ∼20% longer in UKESM1 than in GC3.1. Could this be related?

P19, L8-10: The statement that Mann has a much shorter sulphate aerosol lifetime "due to a combination of lower burden and higher production rates" is backwards. This is just the definition of a shorter lifetime, and does not actually explain why the lifetime is shorter.

P2, L5: The wording "Finally, UKESM1 includes for the first time a representation of a primary marine organic aerosol source" is ambiguous. Previous global atmospheric models have included online estimates of primary marine organic aerosol emissions (e.g. Gantt et al 2015), so this novelty is for the UKESM1 system only. However, this wording is unclear, and makes it sound like UKESM1 is the first global model to include this source. The wording should be modified to make it clear that this is novel only for UKESM1, and not for geoscientific modeling generally.

P10, L16-26: The quantification and comparison to literature for BVOC emissions is insufficient, but seems like it could be easily improved. Please make this comparison quantitative, as "reasonably good agreement" is not a meaningful statement. Estimates of annual global emissions from the comparison studies should be provided. The total monoterpene emissions flux for GC3.1 should also be stated explicitly. Furthermore, it is not clear why isoprene emissions are not quantified.

While the focus of the model is on tropospheric aerosols, one of the advantages discussed of the UKESM1 model is that it incorporates a full stratospheric-tropospheric chemistry mechanism. I was therefore surprised to see no explicit description of how stratospheric aerosols are represented. Is the same GLOMAP scheme used throughout? If so, how are polar stratospheric clouds represented? If not, where and how does the aerosol representation in UKESM1 transition from GLOMAP-mode to some stratospheric scheme? Archibald et al (2019) states that PSC treatments were "recently expanded in UKCA", but that "these improvements did not make it into the UKESM1 version of UKCA discussed here". It would therefore be very useful to have an explicit (if brief) clarification of how stratospheric aerosols are handled, including any transition between schemes.

The authors state that the lack of (eg) nitrate aerosol is part of the reason for underestimation of aerosol mass by the model. However, no description is given in the paper of what components are included in each of the four aerosol "species". An explicit description is needed as to what chemical species are included in each of the four. In particular, the presence or absence of ammonium and nitrate in the "sulphate" aerosol; and what composition is assumed for "sea salt".

Section 2.4.1 states that sea salt emissions are calculated based on Gong (2003). However, this equation is non-linear with respect to surface wind speed, meaning that changes in horizontal resolution can change total emissions. Gong (2003) appears to be assessed using the same setup as Gong et al (2002), in which simulations were performed at a horizontal resolution of $\sim$3.75 degrees compared to the $\sim$1 degree resolution used here. Is any tuning applied to the sea salt emission calculation to account for this? This is particularly important given that marine POA is emitted proportional to sea salt (equation 3 on page 9).

"UKESM1 is believed to be one of the most comprehensively coupled models" (P42 L32) needs to be justified, or it is simply an assertion.

Minor comments, including formatting and typographic errors

P4, L1: ERF not defined on first use.

P7, L1-5: Are all 15 of the competition sub-steps performed together, once per 60-minute time step? Or are 1/3 performed after each 20-minute advection step? Please clarify.

[Figure]

P8, Eq (1): This formatting is confusing and unconventional. I recommend the authors change this to a more conventional stacked format, with the DMS = a form on top and the DMS =b[log. . . form on the bottom.

P10, L1: Commas are incorrectly placed (one is needed after "monoterpenes" and should be removed after "iBVOC")

P16, L9: "diagnositcs" should be "diagnostics"

P20, L18: It seems strange for a burden to be reported in "Tg/yr" – I assume this is a typo.

P32, L2: "simplier" such be "simpler"

P42, L25: "an underestimations" should be "underestimations" (or "an underestimation", possibly)

References

Gantt, B., Johnson, M. S., Crippa, M., Prévôt, A. S. H., and Meskhidze, N.: Implementing marine organic aerosols into the GEOS-Chem model, Geosci. Model Dev., 8, 619–629, https://doi.org/10.5194/gmd-8-619-2015, 2015.

Gong, S.L.: A parameterization of sea-salt aerosol source function for sub- and super-micron particles, Global Biogeochemical Cycles, 2003.

Gong, S. L., L. A. Barrie, and M. Lazare, Canadian Aerosol Module (CAM): A size‐segregated simulation of atmospheric aerosol processes for climate and air quality models: 2. Global sea‐salt aerosol and its budgets, J. Geophys. Res., 107(D24), 4779, doi:10.1029/2001JD002004, 2002.

---

## Author Response (AR1)

**Detailed response to the Editor on the Reviewer Comments on "Description and evaluation of aerosol in UKESM1 and HadGEM3-GC3.1 CMIP6 historical simulations" by Mulcahy et al.**

Dear Editor,

On behalf of all co-authors I would like to thank you for agreeing to handle the above manuscript. We also thank both reviewers for taking the time during this busy period to review the manuscript and for the constructive comments provided. We detail our responses to their specific comments below and feel the revisions made have strengthened and increased the utility of this paper. Furthermore, we also provide detail below on some changes made to the manuscript due to the discovery of a bug in the leaf area index (LAI) ancillary data used in the atmosphere-only (AMIP) UKESM1 simulation.

In the responses below the original reviewer comments are in **bold** while the author comments are in plain text. Changes to the manuscript are place in *italics*.

We hope you find the revised manuscript now suitable for publication in Geoscientific Model Development.

Kind regards,

Jane Mulcahy

**Changes due to LAI bug:**

During the discussion phase, we discovered a bug in the UKESM1-AMIP configuration that is used in some parts of the paper. Essentially, in the atmosphere-only configuration of UKESM1, we prescribe the land surface characteristics using outputs of vegetation fractions, leaf area index (LAI) and canopy heights from the fully coupled UKESM1 model. We discovered that the seasonal cycle of the LAI data was 6 months out of phase. This doesn't have a significant impact on the aerosol simulation assessed here as most of the paper uses the fully coupled historical simulations which do not include this bug. However, in instances where we have used the AMIP simulation (eg: the aerosol and gas phase budget analysis and sensitivity simulations), we have re-run the simulations and re-plotted all affected figures in the manuscript, as well as recalculating the full aerosol budget in Table 4. *No changes to the text were required.*

All of the original data has been retracted from the Earth System Grid Federation (ESGF) and the updated data and an erratum have now been published: https://errata.es-doc.org/static/view.html?uid=5e70c479-9b19-ca91-a9ba-d11febe15377

*Changes made:*

1. **Table 4 budget table has been updated** -no notable impact of bug on budget numbers apart from a small increase in the UKESM1 emissions of monoterpenes from 127 to 130 Tg/yr which leads to a small increase (less than 1 Tg/yr) in the secondary organic aerosol production in the AMIP simulation.
2. **Figure 8 has been updated,** replacing the original UKESM1-AMIP line with the bug-fixed simulation – no notable impact on the AOD timeseries.
3. **Figure 15 has been updated –** no impact on the PMOA emissions**.**
4. **Figure 16 had been updated -** the UKESM1-AMIP and NoPMOA AMIP lines have been updated.

5. **Supplemental Figures 1 and 2 have also been updated** with the new data with small increases found in the monoterpene emissions.

*Response to Reviewer 1*

**Natural emissions: even though the citations leads to detailed descriptions of the emission algorithms, it would be beneficial for the reader to be served mathematical expressions for more of the emissions to easier see how they vary with climate variables such as temperature and wind.**

We don't include the detailed descriptions for all the natural emissions as we feel it would make an already long manuscript too long. In particular, we feel it's not required given the detailed descriptions provided in the cited literature, where implementations deviate from the literature (for example in the case of the DMS seawater parameterization) we document this. We have included a more detailed description of the primary marine organic parameterization as this is implemented into the GLOMAP aerosol scheme for the first time and we feel it's important to describe how the Gantt et al scheme was implemented in the model. There is also a more detailed paper in preparation documenting in detail the dust emissions in UKESM1 and so we do not go into depth on the dust treatment here.

**BVOC: the models only include SOA production from monoterpenes. Although these emissions are scaled up, they are different in nature compared to other BVOCs. A few sentences about the disadvantages of this simplification would be nice.**

We appreciate the reviewer's comment and we recognise that our simplified approach is not without a drawback. However, the impacts of our simplified approach will manifest predominantly at the regional and local scale and will pose less of an issue on the hemispheric and global scale. The most important difference between isoprene and monoterpene emissions are in the geographic distribution. Isoprene is emitted predominantly at tropical and sub-tropical latitudes while monoterpene emissions are located mainly (but not exclusively) at mid- to high latitudes. By scaling the yield from monoterpenes in our SOA scheme we preserve the overall emission magnitude while introducing, to some degree, a bias in the distribution (both isoprenoids and SOA are short-lived and thus not transported very far away from their sources). For the development of this generation of UKESM we deemed the trade-off to be acceptable since UKESM is predominantly a global ESM. Furthermore, the treatment of SOA chemistry in GLOMAP is relatively simple and does not include the degradation of isoprene sources. Further improvements are planned for future versions of UKESM.

We will adapt the text in Section 2.4.4 as follows (new text in red):
*P10 L18:* *"While biogenic isoprene emissions are coupled to the gas-phase chemistry in the UKCA model and thus directly affect tropospheric ozone production and methane lifetime, due to the simple SOA chemical formation mechanism currently employed in the model (Table 2 and 3) only emissions of monoterpenes contribute to the formation of SOA. As already described above the yield of SOA from monoterpene has been doubled from 0.13 used in Mann et al. (2010) to 0.26 here in part to account for missing BVOC sources. This simplified approach preserves the global emission magnitude but may introduce a bias in the geographic distribution of SOA. Isoprene is emitted mainly in the tropics and sub-tropics while the largest sources of monoterpenes are found in the NH boreal regions. The bias will, therefore, manifest on the regional and local scale rather than the global and is expected to be small compared to the large uncertainty associated with modelling BVOC emissions (Arneth el al., 2008)"*

**Oxidation: several of the differences in behavior between the two models are linked to their different treatment of the oxidants and the following differences in oxidation close to and away from emission sources. These statements will benefit from map plots showing horizontal distribution of the oxidants and/or horizontal distribution of loss of precursors through the chemical processes.**

We thank the reviewer for this comment and agree this will make a nice, highly relevant addition to the paper and so in the revised manuscript we have added additional analysis on the oxidants and how the oxidation rates differ between the two models.

We now include a new figure (now Figure 1 in the revised manuscript) showing the surface level distribution of the relevant oxidants, $O_3$, OH, $H_2O_2$, $NO_3$ and $HO_2$ in UKESM1, GC3.1 and their fractional difference. We have extended our analysis of the $SO_4$, $SO_2$ and DMS budgets in Section 5.1 and the evaluation of $SO_4$ surface concentrations in Section 5.2.1 to include reference to the differences in the oxidants, adding the following text:

**P20 L28:** *Surface level oxidants from both models and their fractional differences are shown in Figure 1. Over global oceans, $NO_3$ is significantly lower in UKESM1 and is over 80% lower in the important DMS source region of the Southern Ocean region.*

**P21 L10:** *The timescales for oxidation will be determined largely by the differences in oxidants between the two models (see Figure 1). Apart from $NO_3$ UKESM1 has overall higher oxidant concentrations than GC3.1, particularly for the key chemical oxidants $H_2O_2$, $O_3$ and OH although there are some regions where UKESM1 oxidants are lower.*

**P27 L19:** *Lower concentrations of $O_3$ over Europe (Figure 1c) lead to a nearly 60% lower production of SO4 from aqueous $O_3$ oxidation. The different vertical profile of $SO_4$ production also leads to a shorter dry deposition lifetime of $SO_4$ in UKESM1 (3.7 versus 5.6 days) although this is compensated for by a longer lifetime for wet removal. Regional budget analysis of the $SO_2$ and $SO_4$ budget over North America found very similar oxidation rates and timescales in both models despite some notable differences in oxidant concentrations shown in Figure 1. This supports the comparable performance of both models against the observations across the North American sites and suggests the larger contributing role of emissions and deposition processes (common to both models) to biases in this region.*

**Ice crystals: Cloud droplet number concentration is thoroughly evaluated, but what about the ice crystals? If the model does not include aerosol impact on cold clouds, it should be mentioned. If some of the aerosols can act as INP, that should also be described.**

Aerosols do not act as Ice Nuclei in these models. We have added a statement making this explicit in the current Section 2.8 (will become Section 2.9 in the revised manuscript) "Aerosol-radiation and aerosol-cloud interactions" by adding the following sentence:

**P16 L14**: *"Aerosol-cloud interactions are simulated in warm clouds only and do not act as ice nuclei."*

**Dust aerosol scheme: please add a sentence explaining why dust is not implemented in the regular aerosol scheme, but needs a single aerosol scheme.**

We have amended the following sentence in Section 2.1, **P6 L33:**

*"Mineral dust is simulated in UKESM1 and GC3.1 but not in the modal framework* as the development of the modal dust scheme within HadGEM3 has not yet reached sufficient maturity"

**Cloud droplet number concentration: Can you add some sentences arguing why the satellite datasets are comparable to your model output? Do you use the same criteria for your modeled Nd as in the satellite products (for example cloud fractions > 80 %). With satellites only seeing cloud tops and specific clouds under specific conditions, the model output would be more comparable if the same criteria where used, or if a satellite simulator was applied. If a satellite simulator was applied, it should be mentioned. If not, it should be mentioned that the model output would be more comparable to the satellite products if a satellite simulator was used.**

We agree with the reviewer that the correct sampling of the model data to match the observational criteria is the best way of conducting the comparison. Doing this requires high temporal data from the model (~3 hourly data with some fields being 3-dimensional) in order to sample consistently or a satellite simulator as the reviewer points out. Outputting 3 hourly fields in the fully coupled historical simulations was not possible given the already very large amount of data being output to satisfy the CMIP6 data request. Unfortunately, we do not currently have a satellite simulator capability for cloud droplet number concentrations.

A cloud top CDNC diagnostic (which is a better reflection of what the satellite observes) was available from UKESM1 historical simulations but not GC3.1. Instead we used CDNC at 1km in our comparison. Comparisons of 1km and cloud top Nd in UKESM1 did not show significant differences and so we believe 1km data is a reasonable proxy and it is used from both models in the comparisons here for consistency.  Annual mean climatologies were created from monthly data over the time period covered by the observations.  We exclude data north and south of 60deg in the northern and southern hemispheres respectively where the satellite data is uncertain due to high solar zenith angles and sea-ice. It is worth mentioning that both the Bennartz and Grosvenor datasets have different filtering criterion applied (already documented fully in Section 4.4) and so by including both we hope the encompass the range of uncertainty in the retrievals and hence in the evaluation. We recognise that the detail on how the comparison was conducted was not clear and so in the revised manuscript we have modified Section 4.4 as follows (new text in red):

**P19 L23***: "Annual mean climatologies of simulated Nd at 1km from 2003-2014 are compared with annual means generated from the satellite products. Lack of high temporal outputs from the historical simulations prevents consistent filtering methods from being applied to both satellite and model data. High solar zenith angles and sea-ice are screened for in Grosvenor et al. (2018b) but not in the Bennartz and Rauch (2017) dataset. It is also possible that undetected sea-ice affects the Grosvenor et al. (2018b) dataset. Hence data has been removed, north and south of 60° in the northern and southern hemispheres respectively, in both model and satellite data where retrievals are likely uncertain."*

**The conclusion would benefit from adding a few sentences about which improvement that should be made to future model versions.**

We will add the following to the Conclusions (new text in red):

**P45 L15:** *"The inclusion of marine organic aerosol is furthermore found to improve the seasonal cycle of cloud droplet number concentration in the Southern Ocean although biases associated with the interactive simulation of DMS and Chl-a in UKESM1 are evident. Future model developments will focus on improving these prognostically coupled components and an in-depth evaluation of the*

*chemistry-aerosol coupling will be conducted via detailed evaluation of the complete sulphur cycle including sulphate aerosol production rates. "*

**Technical corrections:**
**p2, l19: "on which cloud droplets can form"!"on which cloud droplets and ice crystals can form"** - Now corrected
**p2, l20: "increase anthropogenic emissions leads to increases" ! "increase anthropogenic emissions usually leads to increases" (the opposite can occur if it's already many aerosols and low supersaturation)** - Now corrected
**p9, Eq. (2): missing two end parentheses.** - Thank you, now corrected.
**p11, Table 2: COS is not explained in the text. Please do so. –** Thank you the Aerosol Chemistry section has now been updated.

**P11 L20: "***In the stratosphere additional sulphur cycle aerosol-chemistry processes are included which are appropriate for non-volcanic sources in the stratosphere (Dhomse et al., 2014; Weisenstein et al., 1997). These include the photolytic and thermal reactions of COS, SO2, SO3 and H2SO4. Reactions of COS and DMS with O(3P) are also included. Volcanic sources of SO2 in the stratosphere are not treated interactively but are specified from a climatology (see Section 2.8).***"**

**p11, Table 2: What happens with MSA? -** MSA is considered inert and is not advected or transported , we have updated the Aerosol Chemistry section to reflect this:

**P11 L14:** "MSA is treated as an inert sink of sulphur and is neither transported nor advected in the model."

**p12, Table 3: be consequent regarding SEC_ORG and Sec_Org** – Now corrected

**p13, l10: add space after . -** Now corrected

**p13, l18, add space after .** - Now corrected

**p13, l24: add space before nm** – Now corrected

**p20, l15: "historican" ! "historical"** - Now corrected

**p21: Table 4 and Table 5 can be merged for easier comparison of the models by using one type of brackets for UKESM, and another type of brackets for HadGEM**

We have now merged the budget values from the 2 models to be in the same Table 4. We have also added a new Table 5 which details the full breakdown of the SO2 budget and production and loss fluxes from each reaction in each model. This provides details on oxidation and deposition loss timescales.

**p22, Figure 1. "mg[SO2]m2" ! "mg[SO2]/m2"** - Now corrected

**p22, l23: remove . Before "for all years"** - Now corrected

**p23, Figure 2: Add one column with differences between the models. –** Now added

**p24, Figure 3: Is the regression correct? It does not look correct for some of the figures**. **The figure will also be better with partly transparent dots. –**

The regression lines aren't shown on the plots but the 2:1, 1:1 and 0.5:1 lines are. We will make this clearer in the caption by adding the following sentence to the caption of Figure 4:

**P26:** *"The 1:1 line is shown in solid black while factor of 2 differences from this line are shown by the dashed grey lines."*

We have also made the dot symbol more transparent – although we found this doesn't make much difference to the look of the plot.

**p26, Figure 5: "netorks" ! "networks"-** now corrected

**p33-34, Figure 10 and 11: add arrows on the colorbar (like the other figures in the paper)** – now corrected

Response to Reviewer #2

**Firstly, the analysis appears to mostly rely on comparison of completed ensembles, with few examples of dedicated simulations which isolate a specific process present in one model but not the other. This makes it difficult to evaluate the true contribution of individual differences between the UKESM1 and GC3.1 to the observed differences. This undermines the authors' claims regarding the value of having a "traceable hierarchy". For example, it is difficult to fully interpret differences in sulphate loading (or to evaluate the accuracy of the authors' interpretations) because the two models use different approaches to distribute sulphur emissions. To truly take advantage of the"traceable hierarchy", more sensitivity simulations – such as the one performed here for evaluation of the effect of the primary marine organic aerosol emissions parameterization – would be needed, and would have significantly improved the paper. Without them, it is difficult to say why exactly one model is more accurate than the other. This mostly constitutes a missed opportunity and a reduction in the paper's utility.**

We thank the reviewer for their comments. While this paper indeed aims to address the important question of how much additional skill in the aerosol simulation is achieved by using a more comprehensive Earth system model we also aim to give an overview of the performance of the fully coupled historical ensembles in order to facilitate understanding of the climate forcing and response associated with aerosols in these models in wider CMIP6 studies. To conduct detailed sensitivities of all aspects we feel is beyond the scope of this paper. We note another paper is in preparation which conducts a comprehensive evaluation using multiple observations of the complete sulphur cycle in UKESM1 and a further paper is in preparation focussing on the dust performance. We agree however in the case of the sulphate differences that more sensitivity analysis of the different treatment of emission height distribution and oxidants would add value to this paper. To this end we have added additional plots and discussion to the revised manuscript detailing the differences in the oxidant distributions between the two simulations and have conducted an additional sensitivity test with the UKESM1-AMIP configuration whereby we change the vertical height distribution of the anthropogenic SO2 emissions to be consistent with GC3.1 specification.

A new figure is included (Figure 1 in the revised manuscript) which shows the surface level distribution of the relevant oxidants, $O_3$, OH, $H_2O_2$, $NO_3$ and $HO_2$ in UKESM1, GC3.1 and their fractional difference. We have extended our analysis of the $SO_4$, $SO_2$ and DMS budgets in Section 5.1 and the evaluation of $SO_4$ surface concentrations in Section 5.2.1 to include reference to the differences in the oxidants, adding the following text:

**P20 L28:** "*Surface level oxidants from both models and their fractional differences are shown in Figure 1. Over global oceans, $NO_3$ is significantly lower in UKESM1 and concentrations are over 80% lower in the important DMS source region of the Southern Ocean.*"

**P21 L10:** *The timescales for oxidation will be determined largely by the differences in oxidants between the two models (see Figure 1). Apart from $NO_3$ UKESM1 has overall higher oxidant concentrations than GC3.1, particularly for the key chemical oxidants $H_2O_2$, $O_3$ and OH although there are some regions where UKESM1 oxidants are lower.*

**P27 L19:** "*Lower concentrations of $O_3$ over Europe (Figure 1c) lead to a nearly 60% lower production of SO4 from aqueous $O_3$ oxidation. The different vertical profile of $SO_4$ production also leads to a shorter dry deposition lifetime of $SO_4$ in UKESM1 (3.7 versus 5.6 days) although this is compensated for by a longer lifetime for wet removal. Regional budget analysis of the $SO_2$ and $SO_4$ budget over North America found very similar oxidation rates and timescales in both models despite some notable differences in oxidant concentrations shown in Figure 1. This supports the comparable performance of both models against the observations across the North American sites and suggests the larger contributing role of emissions and deposition processes (common to both models) to biases in this region.*"

Existing Tables 4 and 5 have been merged to make comparison of the budgets between GC3.1 and UKESM1 easier. We have then added a new Table 5 which details the full breakdown of the $SO_2$ budget including the production and loss terms from the relevant chemical reactions. This allows us to have a more detailed, quantitative analysis of the difference in $SO_2$ and $SO_4$ budgets as well as the $SO_4$ evaluation in the text. We have substantially rewritten large parts of Section 5.1 to add in the additional analysis pertaining to the new figure and new table and resulting insights into the oxidation and deposition loss timescales:

**P20 L4:** "*A full breakdown of the SO2 budget is provided in Table 5.* "

**P21 L6:** "*While the timescales for the oxidation of DMS are longer in UKESM1, the oxidation timescales of $SO_2$ are shorter by 10% (3.7 days compared to 4.3 days, see Table 5). Therefore despite GC3.1 having DMS emissions that are more than double that of UKESM1 and higher global $SO_2$ burdens, the secondary production of $SO_4$ is only 15% higher. The globally shorter oxidation timescales in UKESM1 result in comparable $SO_4$ burdens and contribute to the longer $SO_4$ lifetime in UKESM1. The timescales for oxidation will be determined largely by the differences in oxidants between the two models (see Figure 1). Apart from $NO_3$ UKESM1 has overall higher oxidant concentrations than GC3.1, particularly for the key chemical oxidants $H_2O_2$, $O_3$ and OH although there are some regions where UKESM1 oxidants are lower. Smaller wet scavenging rates also contribute to the longer $SO_4$ lifetime.* "

Furthermore, we conducted an additional sensitivity simulation with the anthropogenic SO2 emission heights specified in the same way as in GC3.1. In Section 5.1 we have added the following text:

**P20 L32:** "*The global burden and lifetime of $SO_2$ is smaller in UKESM1 than in GC3.1. This is driven largely by the different emission injection heights of anthropogenic $SO_2$. Inputting all anthropogenic $SO_2$ at the surface, as is done in UKESM1, leads to higher surface $SO_2$ concentrations close to source regions. This additional $SO_2$ is then more efficiently removed via dry deposition, with dry deposition lifetimes in UKESM1 of 6.6 days compared to 8 days, while wet deposition timescales are longer (14 versus 12 days). A sensitivity simulation, with the emission injection heights for anthropogenic $SO_2$ set-up to be the same as GC3.1, increased the $SO_2$ burden and lifetime to 0.61 Tg and 2.21 days*

*respectively. These values compare better with GC3.1. Notably the simulation did not significantly impact the SO$_4$ budget (see Section 5.2.1). This highlights the important role of the aerosol chemistry and driving oxidants in determining the SO$_4$ budget.*"

Existing Figures 3 and 4 are reproduced in the Supplementary Information section with UKESM1 and the emission height test included. These plots highlight a very small impact of the emission height specification on the SO4 evaluation here. In the SO4 evaluation section (Section 5.2.1) we add two additional paragraphs detailing this additional analysis:

**P27 L1:** "*At most surface measurement sites the difference between UKESM1 and GC3.1 surface SO4 concentrations are generally much less than the difference from the observations with both models exhibiting similar biases. The models generally tend to underestimate the observed surface concentrations except over the western US sites where both models overestimate the observations. One notable exception where the models deviate from one another is over Europe. Here UKESM1 has a consistent negative bias during all years (NMB=-0.25) while GC3.1 is in good agreement with the observations (NMB=-0.03). The similarity between both models is in many ways suprising given the different vertical distribution of the anthropogenic SO2 emissions in both models. In UKESM1 all SO2 emissions are emitted at the surface and therefore one might expect higher surface SO4 concentrations as a result. The SO2 surface concentrations at these sites are higher in UKESM1 (not shown). However as discussed in the previous section most of this excess surface SO2 is efficiently removed by dry deposition (Table 5). A sensitivity simulation was conducted which prescribed the SO2 emission injection heights in the same way as in GC3.1. While this decreased the surface SO2 concentrations to be more comparable with GC3.1 it has only a small impact on the surface SO4 comparison (see Figures 3 and 4 in the Supplement). For example, the NMB at EMEP sites reduces slightly from -0.26 to -0.23 while the correlation coefficient increases from 0.37 to 0.39 compared to NMB and correlation coefficient of -0.03 and 0.44 respectively in GC3.1. Furthermore, differences in DMS emissions will not contribute significantly to the source terms at the measurement sites assessed here. This demonstrates that simulated SO4 production in the key anthropogenic source regions assessed here is oxidant limited rather than SO2 limited.*

*While globally the oxidation timescales of SO2 to SO4 are faster in UKESM1 (see Table 4 and 5) regional analysis of the budget over Europe shows the oxidation timescales are slower by 15% leading to a longer regional lifetime of 1.6 compared to 1.3 days in GC3.1. Lower concentrations of O3 over Europe (Figure 1c) lead to a nearly 60% lower production of SO4 from the aqueous phase oxidation by O3. The different vertical profile of SO4 production also leads to a shorter dry deposition lifetime of SO4 in UKESM1 (3.7 versus 5.6 days) although this is compensated for by a longer lifetime for wet removal. Regional budget analysis of the SO2 and SO4 budget over North America found very similar oxidation rates and timescales in both models despite some notable differences in oxidant concentrations shown in Figure 1. This supports the comparable performance of both models against the observations across the North American sites and suggests the larger contributing role of emissions and deposition processes (common to both models) to biases in this region.*"

**Secondly, the comparisons made between the two models are mostly qualitative and subjective. In spite of an abundance of numerical output, the majority of comparisons made by the authors – both between the two models and against observations – are qualitative. This seems unnecessary, and I strongly recommend that the authors revisit their results and discussion sections to make them more quantitative. Differences are frequently said to be "clear", "small", "large", "high", "low", and so on, with comparisons said to be "good" or "excellent", when for all such cases a quantitative assessment should be straightforward. Statements about agreement are also often (albeit not always) vague and qualitative, such as: "BC and OM emissions are in good agreement with other models" (P19, L34); "Mineral dust emissions initially appear high in UKESM1 compared**

to other models" (P20, L10); "the low bias in the OM concentrations is clearly improved" (P37, L17); "the global dust burden. . . compares well with other models" (P20, L17); and so on. This last example is strange since it was immediately preceded by a statement that comparison is methodologically difficult. The last paragraph of page 20 is similarly problematic, claiming that "BC burdens compare extremely well" but then making no quantitative assessment. Similar statements such as that the model "compares remarkably well with the observations" (P42, L15-16) do not add to the discussion and seem more like assertions than scientific evaluations. Throughout the results and discussions sections, the numerous qualitative comparisons should be replaced with quantitative comparisons. This will enable the readers to evaluate the quality of the comparison for themselves, rather than relying on the authors' opinion. These issues are by no means fatal to the paper, and it is worth noting that there are still many quantitative comparisons made. If the authors can generally replace the more qualitative or subjective comparisons with quantitative ones, it will significantly improve the paper while also maximizing its scientific value. Additional sensitivity simulations to isolate the specific contributions of each individual difference between the two models would also help to elevate the paper.

We appreciate this comment and have gone through the Results section to improve the text to make the evaluation more quantitative across all sections. For example on **P24 L15** we have replaced *"overall both models tend to be negatively biased. UKESM1 underestimates the observations to a greater extent than GC3.1"* with *"both models have an overall negative bias with a normalised mean bias (NMB) of -0.25 in UKESM1 and -0.03 in GC3.1. The larger negative bias in UKESM1 results in a larger root mean square error (RMSE) and lower correlation coefficient compared with GC3.1.* "

We have added comparisons to the AeroCom range where appropriate (recommended in a further comment below) in Section 5.1 and have added analysis on the differences in oxidants and the vertical distribution of SO2 emissions in Sections 5.1 and 5.2 which will further quantify and inform the role of these aspects of the sulphur cycle to the evaluation of sulphate aerosol as already outlined above.

With respect to the specific examples highlighted above
**"BC and OM emissions are in good agreement with other models" (P19, L34)** we will include the emission values from cited literature. We rephrase as:

**P22 L1:** *"BC and OM emissions in UKESM1 (GC3.1) are 9.05 (9.05) and 66.5 (61.6) Tg/yr respectively and are within the range of 7-12 Tg/yr (BC) and 68-123 Tg/yr (OM) reported by other models (Tegen et al., 2019, Textor et al. 2006)."*

**Mineral dust emissions initially appear high in UKESM1 compared to other models" (P20, L10) and "the global dust burden. . . compares well with other models" (P20, L17);**
– we now rephrased this paragraph as follows (new text in red):
**P23 L10:** *"Mineral dust emissions in UKESM1 (7398.5 Tg/yr) are much higher than other models and are more than twice as large as GC3.1 (3102 Tg/yr). The AeroCom dust model inter-comparison reports global dust emissions in the range of 514 to 4313 Tg/yr (Huneeus et al, 2011). However, as dust particles are emitted mainly into the larger bins these heavier particles are rapidly lost through sedimentation leading to an overall short dust lifetime of 0.86 days compared to 1.54 days in GC3.1. The different tuned settings in the CLASSIC dust scheme as well as different soil properties in UKESM1 and GC3.1 lead to the higher global dust emissions in UKESM1. It is noted that due to the structure of the dust code, the dust emission diagnostics include all particles released from the surface, including large particles which are almost immediately re-deposited within a single timestep and so are not transported or interacting with the model in any way. These very short-lived dust particles are also included in the diagnosed deposition values and therefore lifetime. This hampers quantitative comparison of these aspects of the dust lifecycle with other models and observations. However, the*

*global dust burden can be compared, and UKESM1 and GC3.1 differ by only 4 Tg (25%). These global burdens are also within the range of other models which span the range of 6.8 to 30 Tg (Tegen et al., 2019; Huneeus et al. 2011, Textor et al., 2006) with the AeroCom median reported to be 15.8 Tg (Huneeus et al. 2011). These factors will be evaluated in more detail in future studies."*

**"the low bias in the OM concentrations is clearly improved" (P37, L17); - rephrased to:**

*P40 L7: "the low bias in OM concentrations is reduced from Dec to Jun and a positive bias is introduced from Jul to Nov. However, the model now exhibits the correct seasonal cycle and captures the magnitude of the summer peak although the simulated peak emission occurs 2 months too early in the model.*

**P27 L30:** We replace "Simulated BC from both UKESM1 and GC3.1 agree reasonably well with the measurements with a correlation coefficient of 0.4" **with** *"For BC, the correlation coefficient between UKESM1 and GC3.1 and the observations is 0.44 and 0.45 respectively (Figure 6a and b). Both models have very similar RMSE values (0.28 versus 0.27) and have an overall negative bias with GC3.1 exhibiting a slightly larger bias (NMB=-0.23) than UKESM1 (NMB=-0.20)"*

**P28 L6:** We replace "UKESM1 compares well with the observations ……" with "*The RMSE (1.14 for UKESM1 and 1.96 for GC3.1) is higher for OM than for BC in both models and the correlation coefficients are lower at 0.36 and 0.35 for UKESM1 and GC3.1 respectively. Overall, both models are positively biased against the observations. GC3.1 has a much larger positive bias than UKESM1 with a normalized mean bias (NMB) that is 3 times larger than UKESM1 (NMB=0.87 versus NMB=0.24) (Figure 6d)."*

**P32 L27:** We replace "Disparity amongst the observations makes evaluation of AOD over remote oceans difficult" with "*Disparity amongst the satellite observations here makes quantitative evaluation of the simulated AOD over the remote oceans difficult. "*

In some instances, due either to the nature of the observations available (for example campaign based data for N50 and N100 comparisons) or temporal sampling issues (for example with satellite data) we are transparent in the more qualitative nature of the comparison and the associated uncertainties (see P18 L23; P28 L1; P35 L21; P44 L29). In particular, we have paid attention to highlighting uncertainties in observed variables, for example in satellite retrieved AOD products and use multiple products to demonstrate this uncertainty. Too often, model evaluation studies treat observed values as truth without due consideration of observational uncertainty in the analysis. We try to at least address this uncertainty in the evaluation conducted here.

**Other major comments**
**Further to the broader need for quantitative comparisons, several comparisons are made to the AeroCom medians (e.g. P20 L21), but it would be more informative to also include the AeroCom range if possible.**

We will add information on the range where appropriate in Section 5.1.

**P22 L1 :** *"BC and OM emissions in UKESM1 (GC3.1) are 9.05 (9.05) and 66.5 (61.6) Tg/yr respectively and are within the range of 7-12 Tg/yr (BC) and 68-123 Tg/yr (OM) reported by other modelling studies (Tegen et al., 2019, Textor et al. 2006)."*

**P22 L18:** "*Mineral dust emissions in UKESM1 (7398.5 Tg/yr) are much higher than other models and are more than twice as large as GC3.1 (3102 Tg/yr). The AeroCom dust model inter-comparison reports global dust emissions in the range of 514 to 4313Tg/yr (Huneeus et al., 2011).*"

**And with reference to dust burdens on P23 L20:** "*These global burdens are also within the range reported by other models which span the range of 6.8 to 30 Tg (Tegen et al., 2019; Huneeus et al., 2011; Textor et al., 2006) with the AeroCom median reported to be 15.8Tg (Huneeus et al., 2011).*"

**How much of the difference in observed outcomes for sulphate can be explained by the different sulfur dioxide emission approaches? In particular, the lifetime of sulphate aerosol is _20% longer in UKESM1 than in GC3.1. Could this be related?**

We have now added a more detailed analysis of the impact of the different $SO_2$ emission injections height in the paper. We find the impact of the emission injection height, while it changes the surface $SO_2$ concentrations as expected, it has only a very small impact on the $SO_4$ evaluation shown in Figure 4. It also has negligible impact on the lifetime, increasing it slightly from 5.5 to 5.6 days when the emissions are distributed over the same heights as in GC3.1. This adds further evidence that the differences in the $SO_4$ concentrations between the two model is driven by the aerosol chemistry and in particular the oxidation rate differences between the two models. We have added this new analysis to the revised manuscript as outlined above.

**P19, L8-10: The statement that Mann has a much shorter sulphate aerosol lifetime "due to a combination of lower burden and higher production rates" is backwards. This is just the definition of a shorter lifetime, and does not actually explain why the lifetime is shorter.**

You are correct, this is likely due to differences in emissions, prescribed oxidants used and also dynamical model. Mann et al. (2010) used GLOMAP in offline mode driven by reanalysis meteorological fields. This will impact transport and removal processes. We have removed this sentence having realised that we already offer potential explanations as to why the values are different from Mann et al (2010), Textor et al (2006) and Bellouin et al (2013) **on P20 L11 (newly added text in red)**:

"*The differences in lifetime across the AeroCom models and previous GLOMAP configurations reflect the diversity in aerosol processes and aerosol chemistry across the different aerosol schemes but also reflect differences in the host climate models driving processes such as the aerosol tracer transport, water uptake and aerosol removal rates.*"

**P2, L5: The wording "Finally, UKESM1 includes for the first time a representation of a primary marine organic aerosol source" is ambiguous. Previous global atmospheric models have included online estimates of primary marine organic aerosol emissions (e.g. Gantt et al 2015), so this novelty is for the UKESM1 system only. However, this wording is unclear, and makes it sound like UKESM1 is the first global model to include this source. The wording should be modified to make it clear that this is novel only for UKESM1, and not for geoscientific modeling generally.**

Reworded to
**P2 L5:** "*Finally, a new primary marine organic aerosol source is implemented into UKESM1 for the first time.*"

**P10, L16-26: The quantification and comparison to literature for BVOC emissions is insufficient, but seems like it could be easily improved. Please make this comparison quantitative, as "reasonably good agreement" is not a meaningful statement. Estimates of annual global emissions from the**

**comparison studies should be provided. The total monoterpene emissions flux for GC3.1 should also be stated explicitly. Furthermore, it is not clear why isoprene emissions are not quantified**.

We now include the emission range from estimates from the studies cited in the text to make this comparison more quantitative. Thank you for highlighting that we omitted to include the GC3.1 value. We have added the following to section 2.4.4:

*P10 L27:* "*Under present-day conditions iBVOC produces an annual global total monoterpene emission flux of approximately 130 Tg[C]/yr. Global annual total monoterpene emissions are highly uncertain (Arneth et al., 2008) and are poorly constrained by measurements. Past estimates range from 29 Tg(C)/yr to 135 Tg(C)/yr at present-day conditions (Guenther et al., 1995; Arneth et al., 2008, Guenther et al., 2012; Sindelerova et al., 2014; Messina et al., 2016, Hantson et al., 2017). UKESM1 is within the upper range of these estimates. In GC3.1, emissions of monoterpenes are prescribed as monthly averages from the Global Emissions Inventory Activity (GEIA) database which used the Guenther et al. (1995) model. This annual global emission flux is higher than UKESM1 at 137 Tg[C]/yr and is just outside the upper range of previous estimates reported above. The GC3.1 monoterpene emissions are temporally fixed and so do not respond to changes in vegetation or climatic conditions.*"

Isoprene is not quantified as it is not currently used in the SOA chemical formation. We have clarified this in the revised text in Section 2.4.4 as follows (new text in red):

*P10 L18:* "*While biogenic isoprene emissions are coupled to the gas-phase chemistry in the UKCA model and thus directly affect tropospheric ozone production and methane lifetime, due to the simple SOA chemical formation mechanism currently employed in the model (Table 2 and 3) only emissions of monoterpenes contribute to the formation of SOA.*"

**While the focus of the model is on tropospheric aerosols, one of the advantages discussed of the UKESM1 model is that it incorporates a full stratospheric-tropospheric chemistry mechanism. I was therefore surprised to see no explicit description of how stratospheric aerosols are represented. Is the same GLOMAP scheme used throughout? If so, how are polar stratospheric clouds represented? If not, where and how does the aerosol representation in UKESM1 transition from GLOMAP-mode to some stratospheric scheme? Archibald et al (2019) states that PSC treatments were "recently expanded in UKCA", but that "these improvements did not make it into the UKESM1 version of UKCA discussed here". It would therefore be very useful to have an explicit (if brief) clarification of how stratospheric aerosols are handled, including any transition between schemes.**

Stratospheric volcanic aerosols (i.e. produced from explosive volcanic injections of SO2) are represented using the CMIP6 stratospheric aerosol climatology, and this implementation is detailed in Sellar et al. (2020). Briefly, the aerosol optical properties and aerosol surface area density (for heterogeneous chemistry) of these stratospheric aerosols have been imposed in UKESM1 based on the climatology developed by the Swiss Federal Institute of Technology (ETH)  (Thomason et al., 2018). This was based on a combination of satellite observations from the period 1979-2014 and chemical transport modelling (Arfeuille et al., 2014). This means that the mass associated with this aerosol source is not represented explicitly or transported in UKESM1 but the approach ensures much better consistency in stratospheric aerosol radiative forcing and surface-area density between CMIP6 models that use this climatology. Non-volcanic sources of stratospheric aerosol are represented explicitly in UKESM1 based on the UKCA stratospheric chemistry scheme that includes

sulphur cycle aerosol-chemistry processes appropriate to the stratosphere (Dhomse et al. 2014). Advection also facilitates some exchange of aerosol between the troposphere and stratosphere.

We will add the description of the stratospheric aerosol chemistry into Section 2.5 (Aerosol Chemistry) as follows:

**P11 L20:** *"In the stratosphere additional sulphur cycle aerosol-chemistry processes are included which are appropriate for the non-volcanic sources in the stratosphere (Dhomse et al. 2014, Weisenstein et al 2006). These include the photolytic and thermal reactions of COS, $SO_2$, $SO_3$ and $H_2SO_4$. Reactions of COS and DMS with $O(^3P)$ is also included. Volcanic sources of SO2 in the stratosphere are not treated interactively but are specified from a climatology (see Section 2.8)"*

We also add a clarification that the offline oxidant scheme in GC3.1 does not include a representation of stratospheric aerosol:

**P11 L29:** *"The chemistry is only solved below 20 km and so does not explicitly simulate stratospheric aerosol chemistry."*

Section 2.8 referred to above is a new sub-section added to describe the treatment of "Stratospheric volcanic aerosols" as outlined above (**see P15 L28**). We move the existing Section 2.8 to Section 2.9.

We also realised that we accidentally omitted the description of the source of continuously degassing volcanoes in the troposphere, so we have added the following to Section 2.4 (Natural aerosol emissions):

**P8 L10:** *"Emissions of $SO_2$ from continuously degassing volcanoes are represented by the present-day three-dimensional climatology of Dentener et al. (2006). This is a temporally fixed data set with no seasonal variation."*

With respect to polar stratospheric clouds, they are calculated using an equilibrium scheme with sedimentation velocities prescribed for NAT (0.46mm/s) and ice+NAT (17mm/s). As stated above stratospheric aerosol is prescribed using a climatology. Below 12km the interactive GLOMAP-mode aerosols are used. From a chemistry perspective, reactions occur on both PSC types and sulphate aerosols (GLOMAP or climatology). A detailed description and comparison of the "older" scheme employed in UKESM1 and the updated scheme mentioned in Archibald et al (2020) is given in Dennison et al (2019).

**The authors state that the lack of (eg) nitrate aerosol is part of the reason for underestimation of aerosol mass by the model. However, no description is given in the paper of what components are included in each of the four aerosol "species". An explicit description is needed as to what chemical species are included in each of the four. In particular, the presence or absence of ammonium and nitrate in the "sulphate" aerosol; and what composition is assumed for "sea salt".**

GLOMAP-mode carries four chemical tracers for aerosol mass: sulphate, organic carbon, black carbon and sea salt. Whilst the model technically carries the mass of $H_2SO_4$ as the prognostic tracer, it assumes that the sulphate is neutralized to form ammonium sulphate in the troposphere and remains as sulphuric acid in the stratosphere. The scheme as yet does not include nitrate or explicitly model the uptake of ammonia by the sulphate, the latter is simply implicit in the calculation of optical properties and aerosol burden diagnostics. For organic carbon, the model carries the full mass of organic material (including the weight of oxygen and other elements) and assumes a ratio of 1.4 between the mass of the carbon and organic matter where primary emissions of OC are supplied

in units of carbon mass. Black carbon is assumed to be pure elemental carbon, and sea salt is assumed to sodium chloride.

**Section 2.4.1 states that sea salt emissions are calculated based on Gong (2003). However, this equation is non-linear with respect to surface wind speed, meaning that changes in horizontal resolution can change total emissions. Gong (2003) appears to be assessed using the same setup as Gong et al (2002), in which simulations were performed at a horizontal resolution of _3.75 degrees compared to the _1 degree resolution used here. Is any tuning applied to the sea salt emission calculation to account for this? This is particularly important given that marine POA is emitted proportional to sea salt (equation 3 on page 9).**

No explicit tuning of the seasalt scheme has been done and the formulation used is the same as in Gong et al. 2003. Seasalt emissions remain the most uncertain sources of aerosol (Textor et al. 2006).  We agree an assessment of the resolution sensitivity of the scheme would indeed be very interesting. Future work will also investigate the temperature sensitivity of the emissions as demonstrated in Jaeglé et al (2011) for instance.

**"UKESM1 is believed to be one of the most comprehensively coupled models" (P42 L32) needs to be justified, or it is simply an assertion.**

We have rephrased the text to:

**P45 L20:** *"In the development of UKESM1 we consciously worked to ensure as many of the process and cross-component couplings were fully prognostic and interactive as possible allowing the model to simulate a large set of future feedbacks. Based on this we believe UKESM1 is one of the most process and coupling complete ESMs available today."*

**Minor comments, including formatting and typographic errors**
**P4, L1: ERF not defined on first use.** - Now corrected

**P7, L1-5: Are all 15 of the competition sub-steps performed together, once per 60-minute time step? Or are 1/3 performed after each 20-minute advection step? Please clarify.** - The former is correct.

**P8, Eq (1): This formatting is confusing and unconventional. I recommend the authors change this to a more conventional stacked format, with the DMS = a form on top and the DMS =b[log. . . form on the bottom.** - We apologise for the confusion the equations should be stacked vertically, this was an issue with the latex compiler used which we will correct in the revised manuscript.

**P10, L1: Commas are incorrectly placed (one is needed after "monoterpenes" and should be removed after "iBVOC")** - now corrected

**P16, L9: "diagnositcs" should be "diagnostics"** - now corrected

**P20, L18: It seems strange for a burden to be reported in "Tg/yr" – I assume this is a Typo.** - Thank you, this was a typo, now corrected.

**P32, L2: "simplier" such be "simpler"** - now corrected

**P42, L25: "an underestimations" should be "underestimations" (or "an underestimation", possibly)** - now corrected

***Additional References:***

Arfeuille, F., Weisenstein, D., Mack, H., Rozanov, E., Peter, T., & Brönnimann, S. (2014). Volcanic forcing for climate modeling: A new microphysics-based data set covering years 1600-present. Climate of the Past, 10(1), 359–375. https://doi.org/10.5194/cp-10-359-2014

Dennison, F., Keeble, J., Morgenstern, O., Zeng, G., Abraham, N. L., and Yang, X.: Improvements to stratospheric chemistry scheme in the UM-UKCA (v10.7) model: solar cycle and heterogeneous reactions, Geosci. Model Dev., 12, 1227–1239, https://doi.org/10.5194/gmd-12-1227-2019, 2019.

Dentener, F., Kinne, S., Bond, T., Boucher, O., Cofala, J., Generoso, S., Ginoux, P., Gong, S., Hoelzemann, J. J., Ito, A., Marelli, L., Penner, J. E., Putaud, J. P., Textor, C., Schulz, M., van der Werf, G. R., and Wilson, J.: Emissions of primary aerosol and precursor gases in the years 2000 and 1750, prescribed data-sets for AeroCom., Atmos. Chem. Phys., 6, 4321–4344, 2006.

Dhomse SS, KM Emmerson, GW Mann, N Bellouin, KS Carslaw, MP Chipperfield, R Hommel, LA Abraham, PJ Telford, P Braesicke, M Dalvi, CE Johnson, F O'Connor, O Morgenstern, JA Pyle, T Deshler, JM Zawodny, and LW Thomason. Aerosol microphysics simulations of the Mount Pinatubo eruption with the UKCA composition-climate model. Atmospheric Chemistry Physics Discussions, 14:2799–2855, 2014.

Huneeus, N., Schulz, M., Balkanski, Y., Griesfeller, J., Prospero, J., Kinne, S., Bauer, S., Boucher, O., Chin, M., Dentener, F., Diehl, T., Easter, R., Fillmore, D., Ghan, S., Ginoux, P., Grini, A., Horowitz, L., Koch, D., Krol, M. C., Landing, W., Liu, X., Mahowald, N., Miller, R., Morcrette, J.-J., Myhre, G., Penner, J., Perlwitz, J., Stier, P., Takemura, T., and Zender, C. S.: Global dust model intercomparison in AeroCom phase I, Atmos. Chem. Phys., 11, 7781–7816, https://doi.org/10.5194/acp-11-7781-2011, 2011.

Jaeglé, L., Quinn, P.K., Bates, T.S., Alexander, B., and Lin, J.-T., Global distribution of sea-salt aerosols: new constraints from in situ and remote sensing observations, *Atmos. Chem. Phys.* 11, 3137-3157, 2011

Sellar, A. A., Walton, J., Jones, C. G., Wood, R., Abraham, N. L., Andrejczuk, M., et al. (2020). Implementation of U.K. Earth system models for CMIP6. Journal of Advances in Modeling Earth Systems, 12, e2019MS001946. https://doi.org/10.1029/2019MS001946

[revised manuscript text omitted]

 height